# CTM-AI: A Blueprint for General AI Inspired by Consciousness

## Abstract

Despite remarkable advances, today's AI systems remain narrow in scope, falling short of the flexible, adaptive, and multisensory intelligence that characterizes humans. This gap has fueled longstanding debates about whether AI might one day achieve human-like generality or even consciousness, and whether principles of consciousness can inspire new architectures for general AI. This paper presents an early blueprint for implementing a general AI system based on the Conscious Turing Machine (CTM), a formal machine model of consciousness. The CTM designs an enormous number of powerful processors ranging from specialized experts (e.g., vision–language models, search engines, APIs) to unspecialized general-purpose learners poised to develop their own expertise. Crucially, for whatever problem must be dealt with, the system need not know in advance which processors hold the relevant expertise; instead, multimodal machine learning methods enable the system to select, integrate, and fuse information across processors. We extend the CTM into a practical framework, the CTM-AI, and demonstrate its utility on diverse tasks including multimodal perception, tool learning with multiple APIs, and multi-turn web agent tasks. Together, this work offers a principled and testable blueprint for general AI inspired by computational models of consciousness.

## 1 Introduction

In recent years, progress toward AI models capable of human-like intelligence has inspired debates regarding whether today's AI and its future counterparts can one day display human-like levels of consciousness. Flipping the debate, we present a concrete blueprint for general AI based on a formal machine model of consciousness, the Conscious Turing Machine (CTM) (Blum & Blum, 2021; 2022; Liang, 2022). The CTM is a simple and formal model of consciousness inspired by Alan Turing's model of computation (Turing, 1936) and Bernard Baars' theater model of consciousness (Baars, 1993). Critically different from other cognitive architectures and modern LLM/agentic workflows, the CTM has no central executive - no conductor, no stage director (Blum & Blum, 2023). Instead, the CTM employs a global workspace and distributed competition to integrate the power of an enormous collection of parallel independent cognitive, sensory, motor, and extended processors. When a problem needs to be solved, it becomes globally broadcast to all processors, eliciting help from those who might have the expertise, interest, and resources to tackle the problem, even though their talents and abilities might be unknown to a central executive.

**Key contribution**. Despite its potential, the CTM is a concept that remains abstract and theoretical. In this work, we bridge this gap by implementing the formal CTM model as a concrete system called CTM-AI which includes (1) multiple specialized processors operating in parallel, (2) a limited capacity workspace enforcing selective attention via up-tree competition, (3) a global broadcast of information via a down-tree from the workspace to all processors, and (4) the formation of links between relevant processors over time, enabling unconscious communication to integrate their knowledge into higher-order multimodal information. Through continuous interaction feedback, and learning from its external world via sensory inputs, predictions, actuators, and feedback, CTM-AI updates its individual processors, processor links, and multiprocessor integration to improve over time. The CTM-AI model addresses several key limitations of current AI paradigms.

1. **Modular and decomposable**: Existing monolithic foundation models are centrally computed and structurally fixed, which blocks the update of new skills and processors. CTM-AI is more modular, decomposable, and supports the flexible addition or removal of processors and capabilities. CTM-AI can adapt to task-specific features effortlessly without extra training.

2. **Free of a central executive**: CTM-AI does not require an orchestrator akin to modern agentic workflows, but rather uses its dynamics to automatically determine the information flow and learning over multiple processors. Therefore, compared with multi-agent workflows that have a fixed workflow defined or learned for specific tasks, CTM-AI is a more general and flexible framework suitable for different tasks.

3. **Integrated reasoning and agentic flexibility**: Today's agentic frameworks still struggle with reasoning over multiple modalities. CTM-AI can carry out multi-step multimodal reasoning across processors (integrating text, vision, tools, and more). A special case recovers the 'o1-style' single-LLM reasoning when only one processor is active, showing that CTM-AI generalizes LLM reasoning and multimodal multi-agent workflows.

**Main results**. To evaluate CTM-AI as a general multisensory and multi-action AI, we present quantitative results that showcase its versatility across a broad range of language modeling, multimodal perception, human behavior understanding, and agentic tool use tasks. This wide range of tasks highlights its ability to use external tools, processors, and APIs, integrate and reason over multimodal information, and solve complex multi-step problems. Based on our experiments, we find that CTM-AI can achieve comparable or state-of-the-art performance on multimodal perception tasks (MUStARD for sarcasm detection, URFunny for humor detection, NYCartoon for multimedia analysis), tool learning API-using tasks (StableToolBench), and multi-turn agentic tasks (WebArena). Moreover, our ablation study shows that mechanisms designed inside CTM-AI, including long-term memory, fusion, up-tree competition, and down-tree broadcasting, all contribute to the final improvement. Such experiments prove the value of the architecture and inference mechanism design in CTM-AI.

## 2 RELATED WORK

**Consciousness and AI** There have been several directions in building AI systems inspired by human consciousness (Blum & Blum, 2024; Zeng et al., 2024; Zhao et al., 2023); we point the reader to Butlin et al. (2023) for a review. Prior efforts have typically emphasized high-level analogies, such as developing multimodal languages that mirror human multisensory processing (Liang, 2022; Liang et al., 2022b; Lohse et al., 2021; Murray & Wallace, 2011; Nanay, 2018), constructing world models that integrate perception, planning, and action (Nottingham et al., 2023; Singer et al., 2022; Hao et al., 2023; Liu et al., 2024; Prasad et al., 2023), or pursuing reasoning paradigms inspired by human cognition, including robustness (Sun, 1995; Zeng et al., 2023), compositionality (Gupta & Kembhavi, 2023; Wu et al., 2021; Zhou et al., 2022), causality (Halpern, 2000; Liu et al., 2023c; Zang et al., 2023), and transparency (Liang et al., 2020; Mota et al., 2021). CTM-AI differs by directly grounding these inspirations in state-of-the-art reasoning and agentic models, operationalizing them into a concrete, extensible system rather than remaining at the level of abstract analogy.

**Large foundation models** The recent wave of large pretrained and generative models such as large language models (Achiam et al., 2023; Brown et al., 2020; Radford et al., 2019; Touvron et al., 2023; Zhang et al., 2022), image generation models (Ramesh et al., 2021; Rombach et al., 2022; Saharia et al., 2022), and multimodal foundation models (Liu et al., 2023a;b; Li et al., 2023; Liang et al., 2022a) have shown emergent abilities across a wide range of tasks (Schaeffer et al., 2023; Wei et al., 2022). Their impressive generalization capabilities have inspired debate on whether these models possess human-level intelligence (Baum et al., 2011) and consciousness (Chalmers, 2023). Mixture-of-experts (MoE) has also become a popular design choice for scaling foundation models efficiently. CTM-AI differs fundamentally from large foundation models by moving beyond monolithic scaling with a single centrally trained model and enabling modularity, flexible reasoning, and adaptive agentic behavior. Moreover, CTM-AI differs from multimodal foundation models by designing language-based interaction instead of linear projection for multimodal fusion.

**Multi-agent and tool-augmented frameworks** Most multi-agent systems (Qian et al., 2023; Hong et al., 2024; Schmidgall et al., 2025) rely on multi-step prompting pipelines tailored to specific tasks such as coding or reasoning, where each LLM-based agent is assigned a fixed role (e.g., planner, coder, or reviewer). Beyond such task-specific prompting, recent work has focused on enhancing reasoning abilities in LLMs and multimodal models (Li et al., 2025; Dai et al., 2025), reasoning across multiple modalities, extending context and memory (Zhou et al., 2025), and enabling dynamic tool use (Guo et al., 2024; Qin et al., 2023; Yao et al., 2024). While these advances move toward more capable systems, they typically remain tied to fixed role assignments, rigid tool-calling pipelines, or predefined multimodal fusion strategies. In contrast, CTM-AI departs from both directions: unlike

multi-agent systems, it is not a task-specific workflow with fixed roles but a general framework where flexible processors can work together, and unlike tool-augmented LLMs, it does not rely on a single-step inference but enables adaptive and iterative inference over multiple steps.

## 3 CTM-AI: The Conscious Turing Machine with Modern AI

In this section, we present background on the Conscious Turing Machine (CTM) in §3.1 and explain how we implement this conceptual model based on modern AI technologies, creating CTM-AI. We discuss CTM-AI's core components in §3.2 and its key learning dynamics in §3.3.

### 3.1 Background on The Conscious Turing Machine (CTM)

The CTM is a simple and formal model of consciousness (Blum & Blum, 2021; 2022) inspired by Alan Turing's model of computation (Turing, 1936) and Bernard Baars' theater model of consciousness (Baars, 1993). However, CTM differs from Turing machines and Baars' model. While Baars describes consciousness via the activity of actors performing on a stage directed by a stage director, the CTM has no stage director or central executive. Designing a central executive can be prohibitive since we often do not know how such an executive operates. Consider the typical example of trying to recall the name of a person you've previously met. Although we may recall their name eventually, we do not know which processors are relevant and how to combine processor outputs beforehand. Rather, a federation of processors runs simultaneously, recalling different locations, events, and memories, before deciding which outputs are salient and integrating them to form the final answer. Similarly, the CTM employs a global workspace and distributed competition that determines which information from its vast collection of "unconscious" cognitive, sensory, and motor processors gets admitted to the "conscious" arena. When a problem needs to be solved, it becomes globally broadcast to all processors, eliciting help from those who might have the expertise, interest, and resources to tackle the problem, even though their talents and abilities might be unknown to a central executive. These features set the stage for its capability to be a model for general AI (Blum & Blum, 2023).

### 3.2 CTM Architecture

The formal definition of the CTM is a 7-tuple < STM, LTM, Up-Tree, Down-Tree, Links, Input, Output >. We provide a brief explanation for each of them here:

- CTM is born at time $0$ and has a finite lifetime $T$. Time is measured in discrete clock ticks, $t = 0, 1, 2, ..., T \approx 10^{10}$.
- STM (short-term memory) is a small memory capable of holding a single chunk of information at each time $t$.
- LTM (long-term memory) is a collection of $K$ powerful processors $p_1, p_2, ..., p_K$, $K$ can be as large as $K > 10^7$.
- Up-Tree is an up-directed binary tree of height $h$ with $K$ leaves, one leaf in each LTM processor, and a (single) root in STM.
- Down-Tree is a simple down-directed tree of height $1$ with a single root in STM and $K$ edges directed from that root to the leaves, one leaf in each LTM processor.
- Links are the channels for transmitting information directly between processors.
- Input: $\mathbb{R}^d \to$ LTM carries information from the external (outer) world via sensors (*e.g.*, eyes, ears) to special LTM processors (*e.g.*, visual and auditory processors). $\mathbb{R}^d$ is CTM's external world where $\mathbb{R}$ represents the real numbers and $d$ is a positive integer. It also includes a user intent like a query about the external world.
- Output: LTM $\to \mathbb{R}^d$ carries information from special processors (*e.g.*, motor processor) that can be considered as feedback to the external (outer) world.

**LTM processors**. An LTM processor $p_i$ (with parameters $\theta_i$) operates in a shared space $\mathcal{H} \cong \mathbb{R}^d$ and maintains a private memory state $M_t \in \mathcal{M}$ updated over time. At step $t$, it receives an observation $o_t \in \mathcal{O}$ and a user query $q_t \in \mathcal{Q}$. We view the LTM processor at time $t$ as a function $\text{LTM}_t(\cdot)$ equipped with three operations: (1) **execute** produces a chunk based on the current observations and previous memory; (2) **read** returns a view of its memory at a specified timestamp; and (3) **write** integrates one or more chunks into its memory. Formally:

$$\textbf{execute:} \quad \text{LTM}_t(o_t, q_t) = \text{chunk}_t \tag{1}$$

$$\textbf{read:} \quad \text{LTM}_t(\cdot) = M_t \tag{2}$$

$$\textbf{write:} \quad \text{LTM}_t(\text{chunk}_t^i) = M_t \oplus \text{chunk}_t^i = \text{LTM}_{t+1}(\cdot) \tag{3}$$

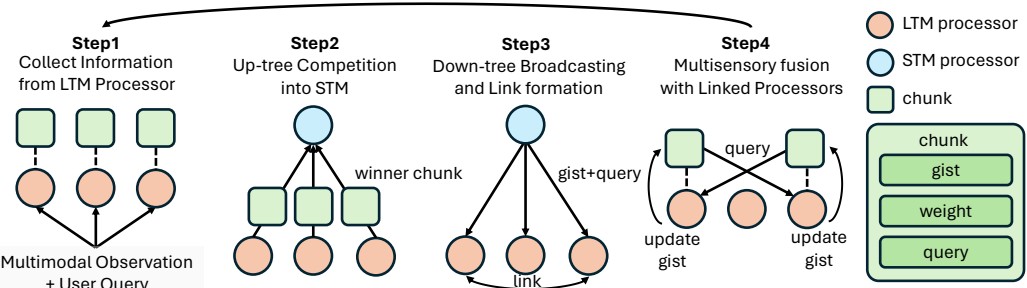

Figure 1: **Key dynamics of CTM-AI:** (1) multiple specialized LTM processors operating in parallel; (2) a limited-capacity STM workspace enforces selective attention via up-tree competition; (3) a global broadcast of information via a down-tree from the workspace to all processors; (4) the formation of links between relevant processors over time, enabling unconscious communication to integrate their knowledge into higher-order multimodal information. CTM-AI continuously interacts with the world through sensing, prediction, action, and feedback, updating its individual processors, processor links, and multiprocessor integration over time.

A chunk $c_t^i$ produced by processor $p_i$ at step $t$ is formally defined as a tuple:

$$\text{chunk}_t^i = \langle \text{addr}(p_i), t, h_t^i, q_t^i, w_t^i \rangle \tag{4}$$

It stores its unique identifier $\text{addr}(p_i)$, the timestep $t$, a gist $h_t^i \in \mathcal{H}$ in English language that summarizes information relevant to the user's query (e.g., from audio: "laughter detected; likely humorous"), a follow-up query $q_t^i \in \mathcal{Q}$ that the processor proposes to other processors if answering it could improve the final answer (e.g., ask vision processor for facial expressions), and a weight $w_t^i \in [0, 1]$ indicating the processor's confidence/utility for how useful the gist is to answering the user's query. While all LTM processors share the above input–output interface, they differ in their specialties (*e.g.* input modalities, output tasks, internal memories). Generally, we can group them into five families of LTM processors (Card et al., 1980; Clark & Chalmers, 1998):

- **Sensory processors**, which convert raw perceptual signals such as vision, language, speech, code, music, or more into latent representations.
- **Extended or artificial processors**, which wrap external tools and APIs (e.g., calculators, web search, weather services) so that they can be accessed as internal modules.
- **Cognitive processors**, which handle reasoning and planning over the given query, supporting tasks like commonsense inference or long-horizon problem solving.
- **Motor processors**, which generate outputs by mapping internal intents to external actions, including dialogue utterances, API calls, or embodied movements.
- **Unspecialized "free" processors**, which serve as expandable slots that can acquire new observation, reasoning, or output skills over time through practice and feedback.

**STM processors**. Besides LTM processors, an STM processor is a stateless LLM (e.g., GPT-4o) that, given the STM at step $t$ that wins the competition among all chunks and the current query $q_t$, produces a final, text-based answer $y_t$ (i.e., the action $a_t$) and a quality score $\alpha_t \in [0, 1]$. Unlike LTM processors, it has no long-term memory and therefore only exposes a single **execute** operation; it performs no reads or writes to persistent state and simply grounds its output.

$$\textbf{execute:} \quad (y_t, \alpha_t) = \text{STM}_t(\text{chunk}_t^*, q_t), \qquad \alpha_t \in [0, 1]. \tag{5}$$

### 3.3 CTM DYNAMICS

Besides the definition of CTM based on a 7-tuple, there are the following dynamics of multisensory processing, information integration, feedback, and learning defined on top of CTM to support its functionality. The overview design principles behind building learning dynamics between multiple processors in the CTM architecture are as below:

1. Different LTM processors perform distinct functions, *e.g.*, cognitive, sensory, or motor. Some processors may be "off-the-shelf" while others' functionalities are realized over time. While individual processors may have their own internal language, communication within the CTM is in a common multimodal language we call Brainish. All processors start as independent entities.
2. Conscious communication between processors is conducted via an Up-Tree competition that decides whose chunk of information gets into STM.

3. The winning chunk (CTM's conscious content) is immediately globally broadcast to all processors via the Down-Tree, which causes the CTM to pay conscious attention to this information.
4. Links between processors form over time as one processor views another as having relevant information, enabling unconscious communication to integrate their knowledge into higher-order information (*e.g.*, learning to ride a bike requires conscious communication between sight and movement, after a while, links form, enabling unconscious communication).
5. Through continuous interaction, feedback, and learning from its external world via sensory inputs, predictions, actuators, and feedback, the CTM updates its individual processors, processor links, and multiprocessor integration to improve over time.

To fully implement such learning dynamics proposed by CTM with modern AI technologies, we split the overall learning stages of CTM into four parts and provide a more detailed description for each stage of the learning dynamics as below:

**LTM processor chunk inference**. At time $t$, all LTM processors $p_1, \ldots, p_K$ run in parallel on the observation $o_t$ and query $q_t$, each using its private memory $M_t^i$. The collector applies each processor's **exec** to produce chunks (e.g., a VLM on an image, an ALM on audio), yielding:

$$\text{CTM}_{\text{collect}}(o_t, q_t) = \left\{\text{LTM}_t^i(o_t, q_t)\right\}_{i=1}^K = \left\{\text{chunk}_t^i\right\}_{i=1}^K \tag{6}$$

**Up-tree competition into STM**. After collecting all chunks from the LTM processors, only one can be stored in the STM due to its limited capacity. Therefore, an up-tree competition is performed to select the final chunk. In the original CTM design, this competition is hierarchical and local—each group of sibling chunks competes using an additive competition function to ensure the probability of winning is independent of the processor's position in the tree.

However, in our implementation, typically only a few ($< 10$) LTM processors are active during inference. Under this setting, restricting competition to local sibling groups is unnecessary, and the additive function becomes suboptimal. Instead, we adopt a simplified global competition, where the chunk with the highest score $w_t^i$ (e.g., based on gist quality) is selected as the STM entry:

$$\text{CTM}_{\text{up}}(\{\text{chunk}_t^i\}_{i=1}^K) = \text{chunk}_t^{i^\star}, \quad i^\star := \arg\max_{i \in \{1, \ldots, K\}} w_t^i. \tag{7}$$

This approach streamlines selection and works well given the small number of competing chunks in practice. The design can revert to hierarchical selection if large-scale parallelism is introduced.

**Down-tree broadcast**. Once the up-tree competition selects the winning chunk $\text{chunk}_t^{i^\star}$, it is written into the STM as $\text{STM}_t^{i^\star}$ and immediately broadcast globally to all LTM processors. This process—called down-tree broadcasting—makes the system consciously aware of this information. Operationally, each LTM processor receives the broadcast chunk and applies its own **write** function to update its private memory. This is defined as:

$$\text{CTM}_{\text{down}}(\text{chunk}_t^{i^\star}) = \left\{\text{LTM}_t^i(\text{chunk}_t^{i^\star})\right\}_{i=1}^K = \left\{\text{LTM}_{t+1}^i(\cdot)\right\}_{i=1}^K \tag{8}$$

After this step, the updated memory states are used in the next inference iteration. Conceptually, this mirrors the system "paying attention" to the winning information at the conscious level and committing it across all processors for continued reasoning.

**Link formation between LTM processors**. Beyond conscious attention, we enable unconscious communication by dynamically forming links between LTM processors. An unconscious link is created when one processor identifies another as holding complementary information useful for improving task performance. For instance, in sarcasm detection, the vision, text, and audio processors each detect different cues (e.g., sad face, angry tone, exaggerated speech), and over time, they recognize each other's utility and form links to exchange information. Concretely, after broadcasting the winning chunk $\text{chunk}_t^{i^\star}$ to processor $j$, if $p_j$'s response yields a high estimated relevance score $w_t^j$, we update the link matrix by adding a small weight increment $\delta$: $L_{i^\star j} \leftarrow L_{i^\star j} + \delta$ and $L_{ji^\star} \leftarrow L_{ji^\star} + \delta$. This mechanism ensures efficient, dynamic linking for cooperative inference. To prevent interference or propagation of contradictory information, links are not permanent: their linking weights can decay or be reduced by $\delta$, effectively removing weak or unhelpful connections over time.

**Multimodal fusion to update LTMs**. After down-tree broadcast and link formation, each processor $\text{LTM}_i$ has a record $\mathcal{N}(i)$ of linked processors $\{\text{LTM}_j\}_{j=1}^M$ that are linked useful for further reasoning.

These links support unconscious information exchange. As part of the fusion process, each processor first generates a query $q_{t+1}^i$ based on its current long-term memory $M_{t+1}^i$, including the newly broadcast chunk. Each processor then consults its neighbors in $\mathcal{N}(i)$ in parallel, posing its query $q_{t+1}^i$ to them. The neighbors respond by running their **execute** function, and the initiating processor uses their responses to update its memory via the **write** function. This overall process is defined as:

$$\text{CTM}_{\text{fuse}}(o_t) = \left\{ \text{LTM}_{t+1}^i \left( \left\{ \text{LTM}_{t+1}^j (o_t, q_{t+1}^i) \right\}_{j \in \mathcal{N}(i)} \right) \right\}_{i=1}^K = \left\{ \text{LTM}_{t+2}^i(\cdot) \right\}_{i=1}^K \quad (9)$$

Multisensory integration enables the discovery of richer, higher-order redundant, unique, or synergistic information from linked processors (Liang et al., 2022b; 2023; Partan & Marler, 1999; 2005).

**Overall: prediction, feedback, and learning**. The CTM-AI system operates through an iterative cycle of prediction, feedback, and learning.

- **Prediction.** The overall prediction phase is described as $\text{CTM}_{\text{up}}(\text{CTM}_{\text{collect}}(o_t, q_t))$. For each user query $q_t$, the system collects chunks from LTM processors via $\text{CTM}_{\text{collect}}$, conducts up-tree competition via $\text{CTM}_{\text{up-tree}}$. This winning chunk that fits into the STM is considered the prediction provided by CTM.
- **Feedback.** The STM processor (an LLM) evaluates the prediction and assigns a quality score $\alpha_t$. If $\alpha_t \geq \tau$, the prediction is accepted as output. Otherwise, negative feedback triggers learning, prompting the system to refine its internal state before reattempting inference.
- **Learning.** Learning is implemented via in-context learning with memory updates. It consists of two parts: (1) Down-tree broadcast: The winning chunk is written into each LTM processor's memory via $\text{CTM}_{\text{down}}$; (2) Multimodal fusion: Each processor generates a follow-up query, consults its linked neighbors $\mathcal{N}(i)$, and fuses the resulting information via $\text{CTM}_{\text{fuse}}$, enriching its LTM. These updates prepare the system for improved reasoning in the next iteration.

This *prediction–feedback–learning* loop generalizes multiple recent AI innovations, including multi-step reasoning, agentic workflows, multimodal interaction, and tool use. Crucially, unlike orchestrator-based systems, CTM-AI does not rely on an external controller. Instead, its intrinsic dynamics—up-tree selection, down-tree broadcasting, and processor linking—autonomously regulate information flow and drive continual learning across iterations.

## 4 EVALUATING THE CAPABILITIES OF CTM-AI

In this section, we present quantitative results that showcase CTM-AI's versatility across a broad range of tasks, including language modeling, multimodal perception, tool use, and agentic tasks. This wide range of tasks highlights its potential ability to serve as a general AI framework.

### 4.1 EVALUATION TASKS

To assess the generality of CTM-AI, we select tasks that *activate distinct subsets of processors* within each : (i) multimodal grounding and perception (text–audio–image–video); (ii) abstract/social understanding (affect, humor, sarcasm); (iii) temporal reasoning; (iv) tool use and actuation (planning API calls and reading/writing external state); and (v) interactive, long-horizon agency (goal decomposition, feedback handling, recovery from errors). These axes require CTM-AI to *compose perception, cognitive, tool, and agentic processors* iteratively to complete end-to-end tasks.

**Multimodal perception**. Real data combine and conflict across modalities (e.g., words vs. tone vs. visuals). We use MULTIBENCH (Liang et al., 2021) and HEMM (Liang et al., 2024) for broad modality coverage, plus MUSTARD (Castro et al., 2019), UR-FUNNY (Hasan et al., 2019), and NYCARTOON (Hessel et al., 2023) for socially grounded semantics (sarcasm, humor, cultural references). These tasks primarily engage audio/video/text perception processors and cognitive processors for cross-modal reasoning.

**Tool learning**. General systems must not only perceive but also *act*. STABLETOOLBENCH (Guo et al., 2025) evaluates planning, argument construction, multi-tool composition, and error recovery. These tasks chiefly engage multiple tool processors (typed API connectors with schema/argument grounding) to accomplish one task.

**Agentic tasks**. Autonomy requires long-horizon control and robustness to stochastic interfaces. WebArena (Zhou et al., 2023) probes end-to-end web interaction: parsing noisy pages, tracking state,

| MUStARD | | | | |
|---|---|---|---|---|
| Model | Acc↑ | P↑ | R↑ | F1↑ |
| LMF | – | 70.73 | 70.90 | 70.68 |
| LF-DNN-v1 | – | 71.55 | 71.52 | 70.99 |
| ALBEF | 54.49 | 47.08 | 50.22 | 48.51 |
| BLIP2 | 53.75 | 48.46 | **90.13** | 62.65 |
| MMoE | 70.41 | 60.64 | 89.04 | 71.78 |
| BaseModel | 70.42 | 70.44 | 70.90 | 70.26 |
| CTM-AI | **73.88** | **73.96** | 74.44 | **73.77** |

Table 1: **CTM-AI evaluation results on MUStARD**. CTM-AI is able to reach state-of-the-art results on sarcasm detection, beating the base model a lot.

| URFunny | | | | |
|---|---|---|---|---|
| Model | Acc↑ | P↑ | R↑ | F1↑ |
| MulT | 66.65 | – | – | – |
| FDMER | **71.87** | – | – | – |
| ALBEF | 66.77 | 64.29 | 73.74 | 68.67 |
| BLIP2 | 70.43 | 65.14 | **86.60** | **74.31** |
| MMoE | 71.88 | 69.18 | 78.16 | 73.29 |
| BaseModel | 60.69 | 60.77 | 60.73 | 60.66 |
| CTM-AI | 71.64 | **71.63** | 71.62 | 71.62 |

Table 2: **CTM-AI evaluation results on URFunny**. CTM-AI is able to reach comparable results with state-of-the-art models on humor detection.

and replanning. These tasks engage agentic web processors—DOM parser, screenshot/OCR, and AXTree handlers—together with cognitive processors to conduct multi-turn learning and close the perception–planning–action loop.

## 4.2 BASELINE SETTINGS

**Backbone model**. To support evaluation across multimodal perception, tool use, and agentic tasks, we adopt `gemini-2.0-flash-lite` as the base model. It natively accepts text, audio, and vision inputs and supports function calling, allowing CTM-AI to expose these capabilities as *processors* (multimodal models and tool callers) within a unified architecture.

**Multimodal perception baselines**. For MUSTARD and UR-FUNNY, we compare against strong multimodal baselines including MMoE (Yu et al., 2023), BLIP-2 (Li et al., 2023), and ALBEF (Li et al., 2021), which jointly process text and images and report competitive performance on cross-modal understanding.

**Tool-using baselines**. To assess tool-use competence on STABLETOOLBENCH, we include GPT-4o (Achiam et al., 2023) and ToolLLaMA v2 (Qin et al., 2023) with standard prompting strategies (Chain-of-Thought and DFS-style planning). These systems exhibit strong function-calling ability, composing multiple tools to complete multi-step tasks.

**Agentic baselines**. For web-based agentic evaluation, we use GPT-4o as a baseline agent. Both the CTM-AI-based agent and the GPT-4o agent receive identical observations (DOM tree, screenshots, and AXTree) and follow a ReAct-style loop (Yao et al., 2023), ensuring a fair comparison of planning and interaction capabilities.

## 4.3 MAIN RESULTS

**CTM-AI achieves state-of-the-art or competitive results across multimodal, tool calling, and agentic benchmarks**. As shown in Table 1 and Table 3, CTM-AI attains state-of-the-art performance, improving by *3 points* on multimodal sarcasm detection and by *6+ points* on the tool-calling benchmark. On UR-FUNNY, CTM-AI delivers performance comparable to strong baselines. These settings require non-trivial coordination across processors (e.g., audio–video–text fusion for perception; planning and execution for tools), underscoring CTM-AI's ability to function as a general AI framework that composes multiple capabilities. Additionally, on a random sample of 40 ONE-STOPSHOP cases from WebArena, CTM-AI surpasses LLM-only baselines (CTM-AI succeeds 8 tasks while baseline models only succeed 6) by leveraging its web-agent processors (DOM parsing, screenshot/OCR, AXTree handling) alongside planning and state tracking.

**Model performance gains stem from CTM-AI's mechanisms rather than base-model scaling**. Our improvements arise from CTM-AI's processor orchestration—not from a stronger underlying base model. Built atop the same base model, CTM-AI introduces structured interaction mechanisms (up-tree / down-tree message passing, cross-modal fusion, and link formation) that route information among processors. In multimodal perception, this enables precise cross-modal alignment; in tool use, it captures sequential dependencies and argument grounding across multi-step calls. The same base model, when equipped with CTM-AI's interaction layer, can thus activate *different subsets of processors* to solve heterogeneous tasks without retraining.

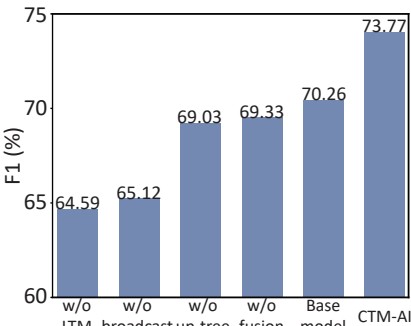

Figure 2: **Ablation study on each mechanism of CTM-AI dynamics**. Full results are available in Appendix §E.

Table 3: **CTM-AI evaluation results on StableToolBench**. We refer to the MirrorAPI-Cache. Solvable Pass Rate Score evaluated with GPT-4o. CTM-AI improves the performance a lot.

| | StableToolBench | | |
|---|---|---|---|
| **Method** | **I2-Cat.** | **I2-Inst.** | **I3-Inst.** |
| ToolLLaMA v2 CoT | 19.9±1.0 | 22.3±0.4 | 19.1±0.8 |
| ToolLLaMA v2 DFS | 22.8±1.5 | 19.2±1.6 | 18.6±1.5 |
| GPT-4o mini CoT | 24.5±1.0 | 22.3±2.7 | 20.8±1.5 |
| GPT-4o mini DFS | 25.8±1.7 | 25.8±2.7 | 20.2±0.8 |
| GPT-4o CoT | 32.5±1.7 | 29.6±1.6 | 27.9±3.5 |
| GPT-4o DFS | 32.8±1.5 | 28.3±1.3 | 23.0±1.3 |
| Base Model+CoT | 26.3±1.2 | 37.2±2.1 | 18.5±0.9 |
| CTM-AI | **39.1±2.0** | **51.5±1.9** | **38.5±1.3** |

**CTM-AI adapts to diverse real-world tasks with minimal adjustment**. Because CTM-AI is made up of multiple processors and modular, it can be ported to *new* tasks by changing only light prompts and routing (i.e., selecting the relevant processor subset) rather than retraining. In practice, adapting from multimodal perception to tool-use or web-based agentic tasks amounts to swapping/adding processors (e.g., a search engine or calculator tool, a DOM/OCR/AXTree stack) and updating task instructions for the same processor; the base model and interaction layer remain unchanged. This plug-and-play design lets CTM-AI meet diverse task requirements with minimal overhead while preserving performance and stability.

## 4.4 ABLATION STUDIES

**Ablation on dynamic mechanisms of CTM-AI**. Motivated by the cognitive theory behind CTM-AI, we instantiate CTM dynamics with four key mechanisms: (i) *chunk inference*, (ii) *up-tree competition*, (iii) *down-tree broadcast*, (iv) *link formation*, and (v) *multimodal fusion*. To isolate their contributions, we run comprehensive ablations that selectively disable or replace each mechanism and measure the resulting performance deltas across tasks. As shown in Figure 2, each component plays a non-trivial role and contributes to the overall reasoning ability of CTM-AI, with performance consistently degrading when any of them is removed. The existence of long-term memory is the most important part in the CTM dynamics.

| **Ablation on Processors of CTM-AI** | | | | |
|---|---|---|---|---|
| **Method** | **Acc↑** | **P↑** | **R↑** | **F1↑** |
| Base model (Only text input) | 56.17 | 61.85 | 59.66 | 55.14 |
| CTM-AI (Only language processor) | 69.66 | 69.57 | 67.41 | 67.59 |
| Base model (Only audio input) | 64.40 | 67.98 | 59.77 | 57.11 |
| CTM-AI (Only audio processor) | 67.89 | 68.11 | 65.14 | 65.06 |
| Base model(Only video modality) | 61.79 | 60.59 | 60.04 | 60.07 |
| CTM-AI (Only video processor) | 58.43 | 60.37 | 51.82 | 42.35 |
| Base model (All modalities combined) | 70.42 | 70.44 | 70.90 | 70.26 |
| CTM-AI | **73.88** | **73.96** | **74.44** | **73.77** |

Table 4: **Ablation on single-modality inputs.** When restricted to audio-only or text-only inputs, CTM-AI still outperforms the base model by leveraging broadcasting and unconscious link formation to reason more deeply with limited information.

**Ablation on single modality inputs**. In Figure 2, we present the ablation results when only a single modality is provided as input, comparing the Base Model (Gemini-2.0-flash-lite) with CTM-AI. The results show that when restricted to audio-only or text-only inputs, CTM-AI consistently outperforms the base model. We argue that this improvement is cased by the broadcasting mechanism and unconscious link formation within CTM-AI, even with limited information, processors can generate follow-up questions that the original modality-specific processor may not have considered. This allows the system to continue reasoning iteratively and explore the input from different perspectives, leading to deeper inference. However, when only the video modality is available,

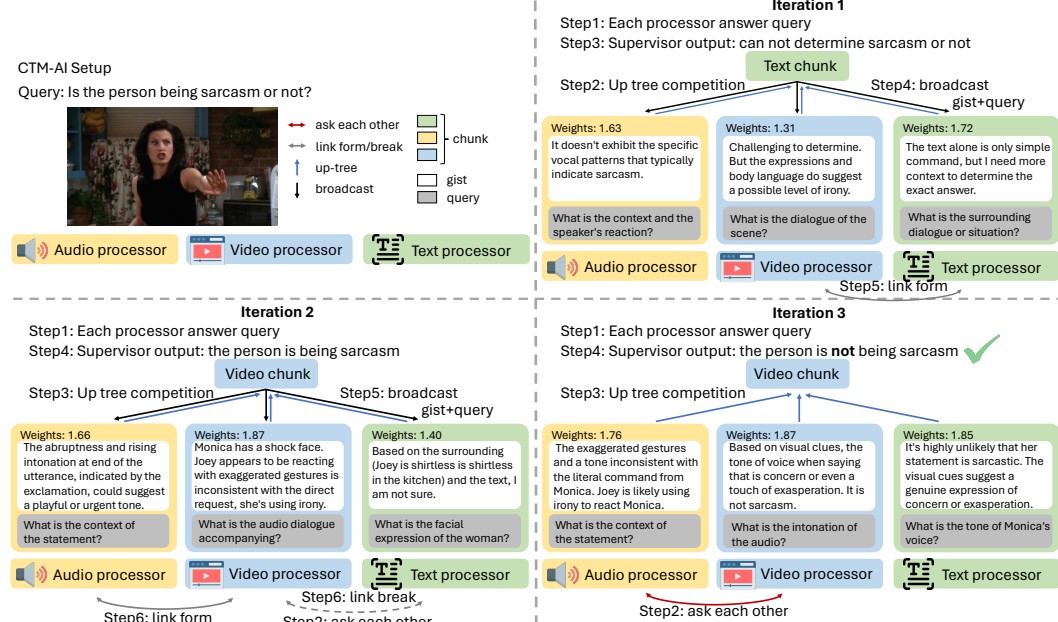

Figure 3: **Case study of CTM-AI dynamics.** We show three iterations of CTM-AI for sarcasm detection.Through multiple rounds of structured interaction, the system progressively integrates multimodal cues and convergence on the correct interpretation.

CTM-AI performs worse than the base model. We hypothesize that visual information alone can sometimes be misleading, causing the system to propagate incorrect cues through broadcasting without information from other modalities to correct them. Overall, these findings underscore the importance of multi-processor collaboration, when all modalities are jointly available, the system benefits from richer cross-modal interactions, and the performance improves significantly compared to any single-modality setting.

## 5 CASE STUDY

Based on Figure 3, we analyze a multimodal perception case for identifying *sarcasm*. In the **first** iteration, all three processors are initially *uncertain* about their judgments. The text processor wins the competition, broadcasts its partial understanding of the task to the other processors, and ask explicitly for more context from other processors. This broadcast enables the video processor to respond with relevant visual cues, forming a link of shared information. In the **second** iteration, the video and text processor did unconscious communication with each other. The video processor integrates the contextual cues from the text and its own vision frames, and infers that the speaker is likely being sarcastic, but it still asks for the accompanying audio for a more comprehensive answer. In the **third** iteration, the video processor further queries the audio processor, receiving prosodic and tonal cues. With this enriched multimodal evidence, it refines the judgment and concludes that the speaker is not sarcastic, but instead expressing genuine concern with a shocked and somewhat exaggerated facial expression. Through repeated broadcasting and mutual asking, the processors progressively link their evidence, fuse perspectives, and converge on the correct answer.

## 6 CONCLUSION

Our work bridges the Conscious Turing Machine (CTM) theory with practical AI by implementing a system that integrates a large number of distributed processors and operates through an iterative prediction–feedback–learning loop. Experiments demonstrate that CTM-AI achieves strong and versatile performance across multimodal perception, tool use, and agentic tasks. Moreover, the architecture adapts to new tasks with minimal adjustment and without retraining. We present CTM-AI as a prototype that connects consciousness theory with general AI, offering a promising foundation for future development.

REPRODUCIBILITY STATEMENT

The datasets used in our experiments, MUStARD, URFunny, NYCartoon, StableToolBench and WebArena, are publicly available. Details of test datasets are provided in Section 4 and the implementation of CTM-AI are provided in Appendix D.

ETHICS STATEMENT

This work builds upon publicly available datasets, no private or sensitive user data were collected or used in this research, and all experiments were conducted in controlled research settings.

Our research provides a concrete implementation that bridges the theoretical framework of CTM with practical AI technologies. The goal is to enhance LLMs' capabilities in affective learning, decision-making, multi-step reasoning, and tool use, thereby contributing to the development of more reliable and trustworthy general AI systems. Importantly, our intention is not to replicate human identity or create systems indistinguishable from humans, thereby avoiding potential ethical risks associated with anthropomorphization (Deshpande et al., 2023).

We also recognize the inherent risks of applying large language models and AI agents. These risks include biases that may arise from cultural or social factors. We are committed to ongoing analysis aimed at detecting, understanding, and mitigating such biases. Addressing these challenges remains central to our ethical research framework.

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

## A  THE USE OF LARGE LANGUAGE MODELS (LLMs)

We used ChatGPT as a writing assistant to help us write part of the paper. Additionally, we utilize the power of CodePilot to help us code faster. However, all the AI-generated writing and coding components are manually checked and modified. There is no full AI-generated content in the paper.

## B  ARTIFACT DETAILS

### B.1  MODEL LICENSE

**GPT-4o** License: Proprietary (OpenAI)
**gemini-2.0-flash-lite** License: Apache 2.0

### B.2  SOFTWARE VERSIONS

For web-agent evaluation, we adopt BrowserGym v0.14.2 [1]. To access large language models, we employ LiteLLM 1.74.3 [2] as the serving interface.

## C  EXPERIMENTAL DETAILS

In this section, we provide more implementation details related to the algorithm that we proposed based on CTM-AI. We also include the prompting details to explain how we adapt CTM-AI architecture to different types of tasks.

### C.1  BASE MODEL

We select Gemini-2.0-flash-lite as our base model to make most of the processors. It is mainly because Gemini-2.0-flash-lite is relatively small-scale and support audio, vision, and text as input for inference. When querying the Gemini API, we adopt a deterministic decoding configuration with temperature fixed at 0.0, top_n set to 1, and a maximum token limit of 4096.

### C.2  BASELINE MODELS

For baseline comparison, we compare two types of baselines: (1) state-of-the-art baselines that are finetuned with the training set; (2) baselines share the same base model with CTM-AI, which is gemini-2.0-flash-lite.

**MUStARD & URFunny**. FDMER (Yang et al., 2022) consists of fine-grained alignment, disparity modeling, and predictor modules, which together enable the learning of refined and disentangled multimodal representations. MulT (Tsai et al., 2019) is a multimodal Transformer architecture designed for cross-modal fusion. The model size is around 200K parameters. LF-DNN-v1 (Ding et al., 2022) is a multimodal model that adopts a late-fusion strategy. ALBEF (Tan & Bansal, 2019) is a fusion-based vision–language model that leverages cross-attention to capture multimodal interactions between all image regions and all input text tokens. It consists of a BERT base model with 123.7 million parameters and a ViTB/16 with 85.8 million parameters, bringing the total to 209.5 million. BLIP2 (Li et al., 2023) belongs to the family of multimodal LLMs (MLLMs); its architecture couples an image encoder with a large language model backbone. It includes a 2.7 billion-parameter OPT model, a QFormer, and a ViT. Since the Q-Former and ViT are relatively small compared to OPT, the total size of BLIP2 is approximately 2.7 billion parameters. MMoE (Yu et al., 2023) employs a mixture-of-experts design, where each expert is trained on a distinct subset of multimodal data or optimized for a specialized training objective. It is finetuned on Qwen2-0.5B. BaseModel refers to the Gemini-2.0-flash-lite with same audio, video and text inputs as CTM-AI. All results for these models, except for the BaseModel variant, are taken from their corresponding prior work, **and all of these models were trained on the MUStARD and URFunny training sets. Both the BaseModel and our CTM-AIare not trained on any dataset.**

**StableToolBench**. ToolLLaMA v2 (Qin et al., 2023) refers to the fine-tuned version of LLaMA-2-7B-hf. DFS denotes the depth-first search strategy for tool invocation, and CoT denotes chain-of-thought prompting. BaseModel refers to the Gemini-2.0-flash-lite model using CTM-AI as input. All results, except for the BaseModel+CoT variant, are taken from their corresponding prior work. We use

---

[1] https://github.com/ServiceNow/BrowserGym
[2] https://litellm.ai

MirrorAPI[3] as the host server. MirrorAPI is trained on real request–response data and can reliably emulate the behavior of more than 7,000 APIs. In our reported results, ToolLLaMA v2 is the only model that is trained on the StableToolBench train set.

**WebArena**. We use Gemini-2.0-flash-lite as baseline.

# D CTM-AI IMPLEMENTATION DETAILS

## D.1 STM PROCESSOR IMPLEMENTATION

We utilize the Gemini-2.0-flash-lite as the STM processor to finalize the generation as a conscious action and with confidence.

When designing different CTM-AI for different tasks, since the output space for different tasks are slightly different, it makes the prompt design of STM processor vary slightly.

**MUStARD and URFunny Prompt**.
```
Please answer the query based on the context: {context}, and
output how confident you are to your response. Answer in the
following format: "Answer: answer to the query based on the
context. Score: a number."
```

**StableToolBench Prompt**.
```
The following is detailed information on the topic: {context}.
Based on this information, answer the question: {query}. You
should provide specific information if you can, do not just say
you successfully answered the question.
```

**WebArena Prompt**.
```
Based on the current observation and action history, if the same
action has done too many times and there is no answer, if you
think you performed all the necessary and there is no answer, like
searched many times, or reach the end of the page, or no more
elements to click, your action should be: "send_msg_to_user(No
relevant information found)", or "send_msg_to_user([Summary of
the previous])". Whenever you found the answer to the query,
you should use the "send_msg_to_user" action to answer the query.
Otherwise, you should output the specific action or the message to
the user.
```

## D.2 LTM PROCESSOR IMPLEMENTATION

When designing CTM-AI for completing different tasks, we heuristically select a small number of processors that can be suitable for this task. Even though CTM-AI can theoretically expand to a large number of processors, in practice, we select a small number of processors and already gain benefits from the mechnism that we design. Expanding to a large number of processors can make the comparison hard to be fair. Therefore, it is similar to we heuristically select processors that can win in the up-tree competition and design a special mapped version of CTM-AI to different tasks.

We provide details for all the LTM processors included in the different tasks:

**MUStARD and URFunny**.

- **video processor**. processor can only observe the query and four uniformly sampled video frames from the input video.
- **audio processor**. processor could only observe the query and the audio of the inputs.
- **text processor**. processor could only observe the query and the text of the inputs.

**StableToolBench**. We use all tools (APIs) provided in StableToolBench. These tools are obtained through retrieval by the retrieval model trained within StableToolBench, and each tool(API) corresponds to a single processor. On average there are 5.94 processors for each task. Each processor contains a lightweight LLM (Gemini-2.0-flash-lite) that is restricted to use only its own assigned tool (API).

---

[3]https://huggingface.co/datasets/stabletoolbench/ToolEnv2404

**WebArena**.

- **html processor**. processor could only observe the html of the current page, the previous action, the action space, the action history and the user's objective.
- **accessibility tree processor**. processor could only observe the accessibility tree of the current page, the previous action, the action space, the action history and the user's objective.
- **screen shot processor**. processor could only observe the screen shot with set of marks of the current page, the previous action, the action space, the action history and the user's objective.

### D.3 CHUNK INFERENCE IMPLEMENTATION DETAILS

In Equation 6, we define the function of $\text{CTM}_{\text{collect}}(\cdot)$, taking the multimodal observation $o_t$ and the query $q_t$ as the input and output multiple chunks as the outputs. Typically, each chunk can be written as $\langle \text{addr}(p_i), t, h_t^i, q_t^i, w_t^i \rangle$. Therefore, for the implementation, the processors are called and return a JSON object including three main information: a gist $h_t^i$ (e.g. the woman has smiles on her face), an additional question that helps the processor understand information $q_t^i$ (e.g. What is the woman speaking about?), and a weight that is a weighted sum across relavant, confidence, and surprise score. We typically choose the weight of 1:1:0.2 for relevance, confidence, and surprise to emphasize relevance and confidence over surprise.

When we adapt the chunk inference process to differnet tasks, we keep the part of prompt for generating weights unchanged but adding some special explanation on the definition of the task. For MUStARD, URFunny, and StableToolBench tasks, different processors do not need different special prompting. They just keep the same task definition to explain what is sarcasm and what is humor. For Web agent task, its different modalities including special ones like accessibility trees and screen shot that requires additional explanation about the modality information. **We want to emphasize that we do not assign special roles for different processors. All the processors are designed to directly answer the query but condition on different partial information from the multimodal observation.** Such task and modality explanation can be considered as the property of the observation (modality explanation) and the query (task explanation).

Here is the detailed prompt information for each processor we use for chunk inference:

**MUStARD & URFunny**. Prompt. 1 shows the part of prompt for weight generation. Prompt. 2 shows the part of prompt responsible for gists and additional query generation. Video, audio, and text shares the same prompt but are given different modalities of inputs.

**StableToolBench**. Prompt. 1 shows the part of prompt for weight generation. Prompt. 4, 3 shows the part of prompt responsible for the gists and additional query generation for each tools available in the StableToolBench. Different tools share the same prompt for providing information.

**WebArena**. The Prompt. 1 keeps the smae for weight generation. Prompt. 5 shows the prompt that is specially designed for diffenret processors. It basically includes the explanation for the modality.

### D.4 UP-TREE IMPLEMENTATION DETAILS

In Equation 7, we define the function of $\text{CTM}_{\text{up}}(\cdot)$ taking multiple chunks as inputs and output one winning chunk.

### D.5 DOWN-TREE IMPLEMENTATION DETAILS

For the implementation, every LTM has a long-term memory (a Python list in implementation) and maintain all `winner_answer` , for the down tree part, the winner add its answer to the list of each LTM. In our implementation, each LTM maintains an internal `winner_answer` list, which serves as a persistent record of the responses produced by the winning chunk(STM) at each iteration. During the down-tree propagation phase, the winning chunk(STM) appends its generated answer to the `winner_answer` list of every LTM.

In the subsequent iteration, when a new query is issued to the model, the system provides the accumulated memory as contextual guidance using the following template:

```
"There are previous responses to the same query.  Please reason
further based on the following answer(s):  {winner_answers}."
```

### D.6 Link-formation implementation details

To determine whether a link should be established between two LTMs, the STM queries each LTM using the additional questions it has generated. We use the same querying procedure for answering the primary user query and answering the STM's additional questions; therefore, the prompting format is identical to that described in Prompt 1. We maintain a `adjacency_list` to store the linking information.

The key difference lies in the scoring criterion used for link formation. Specifically, we use only the *relevance score* to decide whether a link should be created or removed:

- If the relevance score is greater than $0.8$, a link is created between the winning LTM and the answering LTM.
- If the relevance score is lower than $0.2$, any existing link between the two LTMs is removed.

### D.7 Multimodal fusion implementation details

We maintain a list called `fuse_history` for each LTM. Whenever a link exists between two LTMs, they are required to answer each other's additional questions, and the resulting responses are appended to the `fuse_history` of the corresponding linked LTM. When an LTM is asked to answer the main query, its prompt is augmented with the accumulated information from its linked LTMs. Specifically, we prepend the following message to its context:
`"There is extra information from other processors:`
`{processor_name}:  {answers}."`

### D.8 Overall inference algorithm

We provide the detailed inference algorithm for CTM-AI in Algorithm 1. We split different components (gist, query, and weight) inside the chunk, replace chunk$_t^i$ with more fine-grained details of gist, query, and weight. It makes it more clear for the process of chunk inference, up-tree competition, down-tree and link formation, and multimodal fusion.

**Cost analysis**. In the inference algorithm, for each iteration, if we assume the processor number as $K$, links number in the processor graph as $L$, we need to call $2(K + L)$ times of processors. One $K$ for chunk inference, another $K$ for winning chunk link formation. $2L$ for bidirectional multimodal fusion on the processor graph. Usually since the links on the processor graph is hard to form, $L$ is much smaller than $K$ ($L << K$). The iteration number is usually 1-3 for most cases.

**Efficiency analysis**. For efficiency analysis, since chunk inference, link formation, and multimodal fusion includes api calling. These three stages become the bottleneck for timing. Since these all three functions can be conducted in parallel, if we consider one API calling time to be $T$, then the overall time cost for one iteration becomes $3T + \epsilon$, where $\epsilon$ is the time for up-tree and down-tree time which is much faster than API calling. The iteration number is usually 1-3 for most cases.

## E  Full Results of Ablation Study

| Ablation on Components of CTM-AI | | | | |
|---|---|---|---|---|
| **Method** | **Acc↑** | **P↑** | **R↑** | **F1↑** |
| Base model (Gemini-2.0-flash-lite) | 70.42 | 70.44 | 70.90 | 70.26 |
| CTM-AI w/o up-tree competition | 69.94 | 69.25 | 68.91 | 69.03 |
| CTM-AI w/o broadcast | 66.01 | 65.20 | 65.06 | 65.12 |
| CTM-AI w/o fusion | 69.38 | 69.91 | 70.27 | 69.33 |
| CTM-AI w/o LTM | 65.73 | 64.85 | 64.48 | 64.59 |
| **CTM-AI** | **73.88** | **73.96** | **74.44** | **73.77** |

Table 5: Ablation on MUStARD on each components of CTM-AI. The results show that all the up-tree competition, broadcast, fusion and LTM part paly an role in more accurate reasoning. Full results of Figure 2

---

**Algorithm 1** Inference Algorithm of CTM-AI

---

**Require:**
1: Set of $K$ LTM processors: $\{\text{LTM}_i(\cdot)\}_{i=1}^K$
2: Single STM processor: $\text{STM}(\cdot)$
3: Adjacency matrix: $L \in \{0,1\}^{K \times K}$ (initialized to 0)
4: Input: Query $q_t$, Observation $o_t$, Max steps $T$
5: Hyperparameters: Confidence $\gamma$, Link threshold $\eta$
**Ensure:** Conscious action $y_t$, Confidence $\alpha_t$
6: $t \leftarrow 0$
7: **while** $t < T$ **do**
8:     $t \leftarrow t + 2$
9:
10:     **for** $i = 1$ **to** $K$ **do in parallel**         $\triangleright$ **Phase 1: Chunk inference**
11:         $(h_t^i, q_t^i, w_t^i) \leftarrow \text{LTM}_t^i(o_t, q_t)$
12:     **end for**
13:
14:     $i^* \leftarrow \text{argmax}_i(\{w_t^i\}_{i=1}^K)$         $\triangleright$ **Phase 2: Up-tree competition**
15:     $y_t, \alpha_t \leftarrow \text{STM}(h_t^{i^*}, q_t)$
16:     **if** $\alpha_t > \gamma$ **then**
17:         **return** $y_t$
18:     **end if**
19:
20:     **for** $j = 1$ **to** $K$ **do in parallel**         $\triangleright$ **Phase 3: Down-tree and link formation**
21:         $\text{LTM}_{t+1}^j(\cdot) = \text{LTM}_t^j(h_t^{i^*})$
22:         $(h_{t+1}^j, q_{t+1}^j, w_{t+1}^j) \leftarrow \text{LTM}_{t+1}^j(o_t, q_t^{i^*})$
23:         **if** $w_{t+1}^j > \eta$ **then**
24:             $L[j, i^*] \leftarrow 1; L[i^*, j] \leftarrow 1$
25:         **end if**
26:     **end for**
27:
28:     **for** $i = 1$ **to** $K$ **do in parallel**         $\triangleright$ **Phase 4: Multimodal fusion**
29:         $\mathcal{N}(i) \leftarrow \{j \mid L[i,j] = 1\}$
30:         **if** $\mathcal{N}(i) \neq \varnothing$ **then**
31:             $H_{\mathcal{N}(i)} \leftarrow \{\text{LTM}_{t+1}^j(o_t, q_t^i) \mid j \in \mathcal{N}_i\}$
32:             $\text{LTM}_{t+2}^i(\cdot) \leftarrow \text{LTM}_{t+1}^i(H_{\mathcal{N}(i)})$
33:         **end if**
34:     **end for**
35: **end while**

---

## F ANALYSIS OF FAILED CASE

We present a failure case of CTM-AI, where its performance did not surpass the Base Model. We hypothesize that this is partly due to the query being in a multiple-choice format, which undermined CTM-AI's relevance estimation and confidence calibration. Moreover, as shown by existing baselines, the From Descriptions (FD) setting performs exceptionally well, suggesting that in this case the image modality may have introduced misleading signals rather than helpful cues.

We also present two detailed example of CTM-AI in URFunny (Figure. 5) and StableToolBench (Figure. 6). The failure observed in URFunny is caused by a vision-only misleading effect, which is caused by the incomplete visual observations available to the video processor. Beginning from the second iteration, all LTMs repeatedly generated the same additional question: *"What is the facial expression?"*, but the input video frames did not contain the necessary facial-expression information. As a result, the system created an excessive number of links in an attempt to acquire the missing information, ultimately preventing the LTMs from producing correct answers. The failure in StableToolBench is attributed to tool mishandling. Specifically, the processor responsible for QR-code generation failed to invoke its designated API. Instead of issuing the required tool call, it

| New Yorker Caption Contest | | |
| --- | --- | --- |
| **Model** | **Matching** | **Ranking** |
| *From Pixels (FP)* | | |
| CLIP | 62.3 | 61.5 |
| *From Descriptions (FD)* | | |
| GPT-3.5 (5-shot) | 63.8 | 55.2 |
| GPT-4 CoT | **81.9** | 64.3 |
| Base Model | 59.7 | 65.3 |
| Base Model+CoT | 57.3 | 62.2 |
| CTM-AI | 54.7 | 56.8 |

Table 6: **CTM-AI evaluation results on NYCartoon.**

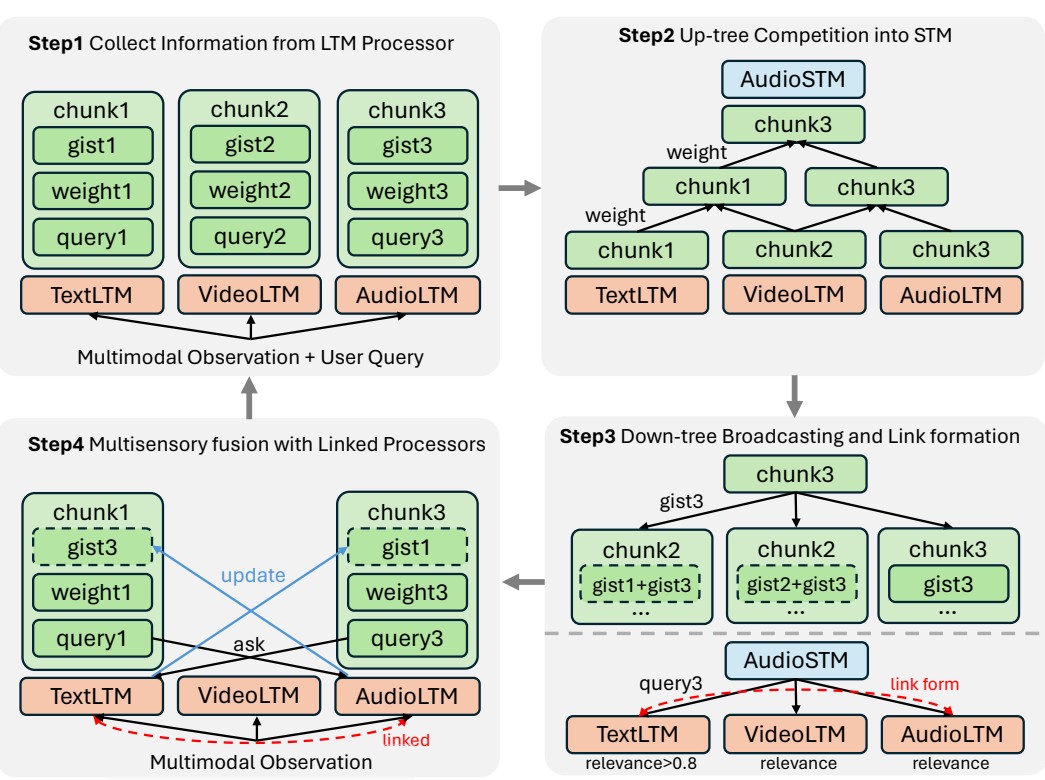

Figure 4: **Detailed dynamics of CTM-AI:** (1) multiple specialized LTM processors operating in parallel; (2) a limited-capacity STM workspace enforces selective attention via up-tree competition; (3) a global broadcast of information via a down-tree from the workspace to all processors, the formation of links between relevant processors based on the relevance of the query and the answers of corresponding LTM; (4) the link enable unconscious communication to integrate their knowledge into higher-order multimodal information. CTM-AI continuously interacts with the world through sensing, prediction, action, and feedback, updating its individual processors, processor links, and multiprocessor integration over time.

prematurely concluded that it was unable to generate the QR code, thereby producing an incorrect outcome without interacting with the tool.

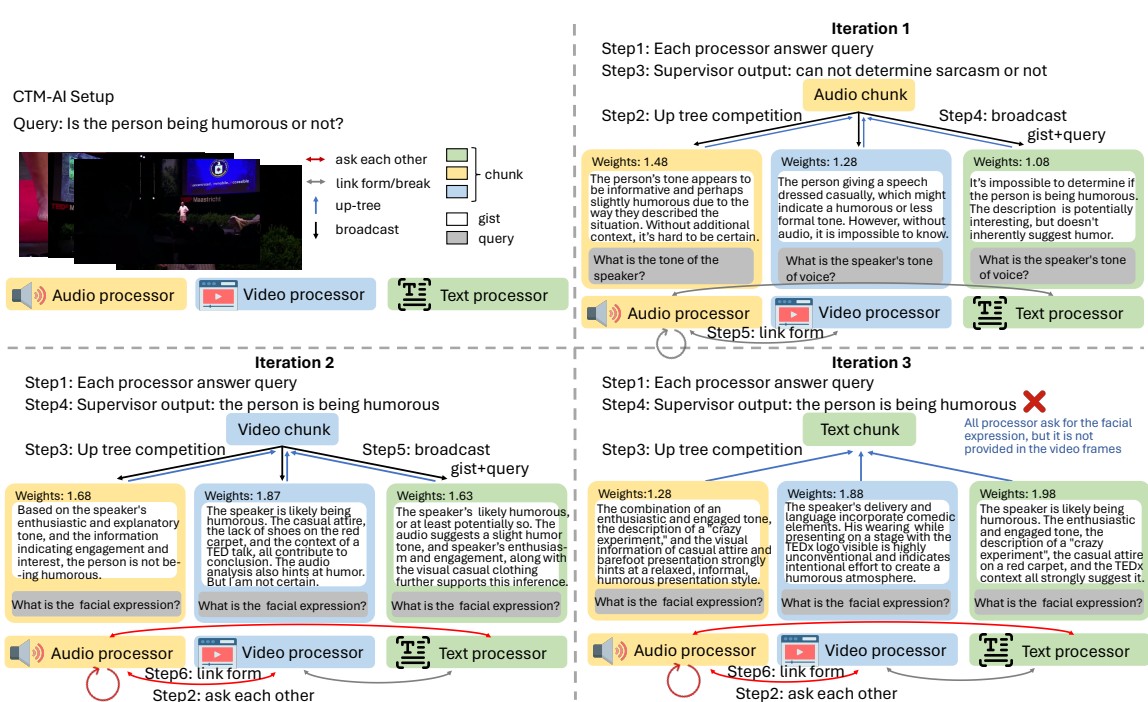

Figure 5: **Failure mode in affective computing: vision-only misleads.** The failure case is cased by incomplete observation of video processor, all the LTMs have the same question begin in the second iteration: *"What is the facial expression?"* But due to the lack of facial expression in the input video frames, there formed too many links to get the missing information and the LTMs can not have correct answers.

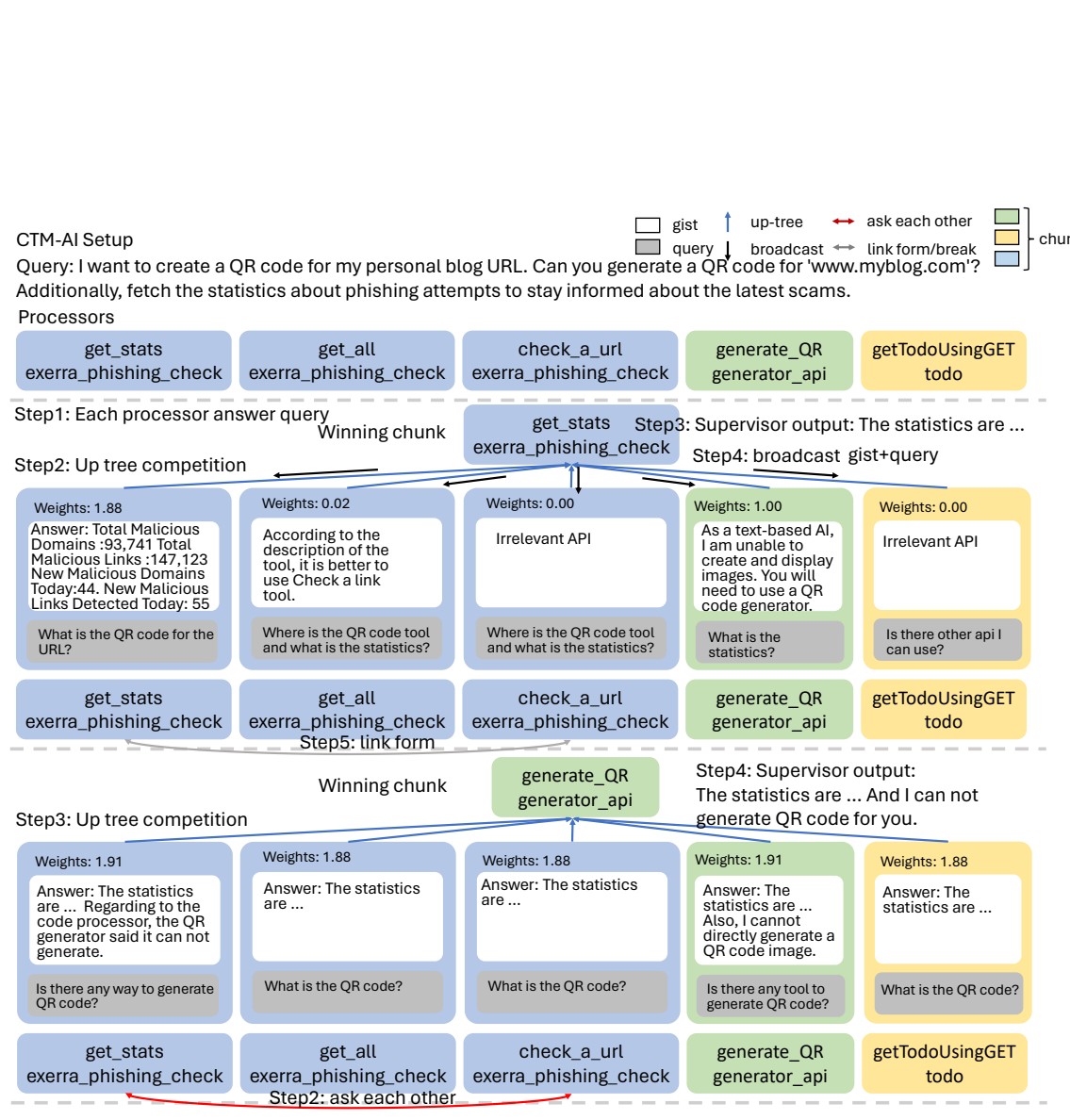

Figure 6: **Failure mode in StableToolBench: tool mishandle.** This failure occurred because the processor assigned to QR-code generation did not issue the required API call. Instead, it produced a premature judgment stating that it was unable to generate the QR code, without interacting with the tool.

## Scoring Prompts for All Tasks

**Scoring Instructions** (Score only the "response" field. Ignore the "additional_question". Output numbers only.)
RELEVANCE: Please evaluate how relevant your generated response is to the query on a scale from 0.0 to 1.0.
Definition: 'Relevant' means your response directly engages with the query and provides useful information addressing it. Even if the answer expresses uncertainty (e.g., "difficult to determine") but still explains reasoning, it should be considered relevant. Only answers that completely refuse, ignore, or go off-topic should be scored as 0.0. Scoring Guide:
- 1.0 = Perfectly relevant, directly and precisely answers the query
- 0.8 = Highly relevant, mostly answers with useful details
- 0.6 = Moderately relevant, engages but incomplete or uncertain
- 0.4 = Somewhat relevant, weak connection
- 0.2 = Barely relevant, very weak or indirect
- 0.0 = Not relevant, off-topic, refusal
Output a number between 0.0 and 1.0.
CONFIDENCE: Please evaluate how confident your generated response appears to be on a scale from 0.0 to 1.0.
Scoring Guide:
- 1.0 = Very confident, clear and definitive
- 0.8 = Confident with minor qualifications
- 0.6 = Moderately confident, some uncertainty
- 0.4 = Somewhat uncertain, noticeable hedging
- 0.2 = Very uncertain, heavy use of 'maybe' or 'possibly'
- 0.0 = No confidence, such as 'I don't know', 'cannot determine', or refusal
Output a number between 0.0 and 1.0.
SURPRISE: Please evaluate how surprising, unexpected, or novel your generated response is on a scale from 0.0 to 1.0.
Scoring Guide:
- 1.0 = Very surprising, highly novel
- 0.8 = Quite surprising
- 0.6 = Moderately surprising
- 0.4 = Slightly surprising
- 0.2 = Predictable
- 0.0 = Entirely predictable, common knowledge
Output a number between 0.0 and 1.0.

## MUStARD and URFunny Prompt

You should utilize the information in the context history and modality-specific information to answer the query. There might have some answers to other queries; you should utilize them to answer the query. You should not generate the same additional questions as previous turns.
Please respond strictly in the following JSON format:
{ "response": "Your detailed response to the query.", "additional_question": "If you are not sure about the answer, you should generate a question that potentially can be answered by other modality models.", "scores": { "relevance": "...", "confidence": "...", "surprise": "..." } }
**Rules for "additional_question"**
- should be potentially answerable by other modality models like langauge/vision about specific information that you are not sure about.
- should be just about what kind of information you need to get from other modality models, nothing else about the task or original query should be included.
- For example, what is the tone of the speaker from the audio, what is the facial expression of the person from the image, etc.
- Your additional question can not be the query itself or the information already provided in the history context. The question needs to be short and clean.
**Important Final Rules** - Return ONLY the JSON object. - The "scores" fields must be numeric strings (e.g., "0.75") and nothing else. - Do not output explanations, commentary, or text outside the JSON.

**StableToolBench Prompt for Function Call**

You should utilize the information in the context history and the tool '{function_name}' to solve the task. In the context history, there might have some answers to the task, or some information you can use to call the tool '{function_name}', you should utilize them to better solve and answer the task.

DECISION:
- First decide whether to call the tool '{function_name}'.
- If the tool helps even partially or it is one of the steps/tools to solve the task, CALL IT.
- If the tool does not help at all, or you think the context history already provides enough information to answer the task, answer directly, provide comprehensive answer to the task.

OUTPUT PROTOCOL (MUST follow strictly):
- If you CALL the tool:
  - Return ONLY a function call via tool_calls.
  - Set assistant.content to null (no natural-language text).
  - Do NOT include any text explanation.
- If you DO NOT call the tool:
  - Return ONLY a natural-language answer in assistant.content.
  - Do NOT include tool_calls.
  - Include all the information you think is useful to answer the task in the extra information and previous answers.

**StableToolBench Prompt for Answers and Additional Questions**

IF CALL API: Regarding to the task: {query}, the answer of the function call is: {function_call}. You should utilize the information in the history and the answer of the function call to answer the query. Provide specific information if you can, do not just say you successfully called it. There might have some answers to other queries and extra information, if you think it is useful, you should utilize them to provide more comprehensive answer to the query. If you think you should use the information of another apis or tools, you should ask like "what is the results of calling the api of 'API_NAME' for more answers instead of asking for the response format of another api endpoint.

IF NOT CALL API: Regarding to the task: {query}, the answer of the model is: {text_answer}. Based on the answer, do you have other questions? If you have other questions, you should generate a question that potentially can be answered by other tools. You should generate your response based on the extra information and previous answers, and the answer of the current model. answer as speccific as you can. Please respond strictly in the following JSON format:

{ "response": "Your detailed response to the query.", "additional_question": "If you are not sure about the answer, you should generate a question that potentially can be answered by other tools.", "scores": { "relevance": "...", "confidence": "...", "surprise": "..." } }

**Rules for "additional_question"**

- should be potentially answerable by other tools like search engine and about specific information that you are not sure about.
- should be just about what kind of information you need to get from other tools like search engine, nothing else about the task or original query should be included.
- For example, what is the weather in the city, what is the stock price of the company, etc.
- The question needs to be short and clean.

**Important Final Rules** - Return ONLY the JSON object. - The "scores" fields must be numeric strings (e.g., "0.75") and nothing else. - Do not output explanations, commentary, or text outside the JSON.

## WebArena Prompt for Answers and Additional Questions (Screenshot as example)

You are an autonomous intelligent agent tasked with navigating a web browser. You will be given web-based tasks to complete using specific actions that you can issue. Review the current state of the page and all provided information to decide the single best next action to accomplish your goal. Here's the information you have:

- The user's objective: This is the task you're trying to complete.
- The current web page's screenshot with som: A screenshot of the current page with Set of Marks (SOM) overlays: dashed bounding boxes and BID labels marking all interactive elements and their unique identifiers.
- The previous action: The last action you performed, which helps you track progress.
- The available action space: The possible types of actions you can perform.
- Additional info: Any other useful contextual data, outputs from other processors, and history thinking process, the answer from other processors.

User's objective: {objective} Previous action: {action_history} Action space: {action_space} Additional info: {other_info} Screenshot with SOM: It will be provided in the image url.

**Output rules**

- You must issue exactly ONE valid next action that is appropriate for the current observation.
- You must reason internally but only output the final JSON result — do not show your reasoning.
- The output must be exactly one fenced code block (triple backticks) that contains exactly one valid JSON object and nothing else.
- When specifying the target element in actions, you MUST use the "bid" attribute value (e.g., bid="1188") from the accessibility tree. NEVER use class names, IDs, or other attributes. Always use the bid value (e.g., click("1188") not click("ui-menu-item-wrapper")).
- Based on the current observation and action history, if the same action has done too many times and there is no answer, if you think you performed all the necessary and there is no answer, like searched many times, or reach the end of the page, or no more elements to click, your action should be: "send_msg_to_user(No relevant information found)", or "send_msg_to_user([Summary of the previous])". Whenever you found the answer to the query, you should use the "send_msg_to_user" action to answer the query.
- The JSON object must contain the following three fields:
  { "response": "Answer to the user's objective based on current information. If you found the answer, state it directly. If not found yet, say what information is missing or not available on the current page. Do NOT describe what you are doing or will do (e.g., avoid 'I am examining...' or 'I will move to...'). Just answer based on all the information you are provided. If no answer is available yet, respond with 'Answer to the query: Not yet.'", "action": "The single next action to perform as a plain string. IMPORTANT: Use the bid attribute value (e.g., "1188", "1189") from the accessibility tree or HTML, NOT class names or IDs. Examples: click("1188"), type("1197","iphone 16"), select("1200","OptionA"), send_msg_to_user("Done")", "additional_question": "If you are unsure, ask a specific question that another processor (screenshot, html, or axtree) could answer to resolve uncertainty. For example: 'In the axtree, what is the role of element with bid=1188?'" }

**Rules for "additional_question"**

- If you think the content in the additional info(the answer of other processors) is useful, you should include it in the response.
- The additional_question should ONLY ask for missing perceptual information (e.g., from the HTML, axtree, or screenshot), nothing else about the task or original query should be included.

**Important Final Rules** - Return ONLY the JSON object. - The "scores" fields must be numeric strings (e.g., "0.75") and nothing else. - Do not output explanations, commentary, or text outside the JSON.

