# OpenReview forum: "CTM-AI: A Blueprint for General AI Inspired by Consciousness"
_ICLR.cc/2026/Conference — Submitted to ICLR 2026_

### Official Review · Reviewer_NwUJ · 2025-10-28

**Soundness:** 1
**Presentation:** 3
**Contribution:** 2
**Rating:** 2
**Confidence:** 3

**Summary:**

The paper proposes a flexible multi-modal agentic framework inspired by Blum&Blum's Conscious Turing Machine (CTM). Many parallel processors (e.g., LLMs, VLMs, ALMs, ...) run in parallel. At every step, one output (or 'gist') is chosen based on a self-assigned weight, written into a global workspace, and broadcast to all processors. In addition, 'subconscious' links can be formed between processors, which allows them to answer each others queries. A supervisor model decides the output from the information in the global workspace. Experiments show that such agents are competitive with baselines, or even outperform them. All components (parallel sub-processes, global broadcasting, competitive gist selection, link-forming) contribute to the overall performance.

**Strengths:**

- The framework is interesting and may have the potential to be a general setup for agentic systems.
- Taking inspiration from the Conscious Turing Machine opens up a fascinating perspective on the parallels (and differences) between the human mind and AIs.
- In the experiments, the CTM-AI performs competitively with, or even outperforms, several baselines on a range of multi-modal tasks.
- An ablation shows that all components contribute to the CTM-AI's performance.

**Weaknesses:**

The most important weak point is lack of detail in the implementation and experimental protocol. Agentic systems are difficult to reliably evaluate. It's all the more important to be very clear and explicit about the exact setup of the system and the experiments. The paper gives barely any detail on either, which is the main reason why I vote reject at the moment. Another concern is the relationship between the proposed framework and the CTM, as well as consciousness in general.

**Questions:**

Below I list important points that, when addressed in a satisfactory way, would improve my score:
- What are the exact prompts used for the different processors? How many different prompts were tried before settling on the ones for used in the experiments? How does this compare to the baselines? The paper claims that only very superficial changes are necessary to adapt it to different tasks. So were the same prompts (with only minor adaptations) were used in all experiments?
- How many, and which kind of LTM processors (what exact inputs, which modalities, and which models) were used in the different experiments?
- How exactly are the weights assigned to the gists? How exactly are links between LTM processors established?
- How much compute (how many tokens, how many FLOPs) is used for each of the experiments? How does this compare to the baselines?

Other things I would wished to be addressed are the following:
- CMT-AI's placement in the literature seems to me a bit superficial. It is surely not the first work proposing a self-organized agentic system that uses natural language as a common medium of communication between specialized agents. As one example, "Mindstorms in Natural Language-Based Societies of Mind" (Zhuge et al., 2023) proposes related concepts.
- The algorithm in the appendix at the moment contains many functions which are not clearly defined, which strongly limits is usefulness. How do Fuse(), UpTree(), Act() and IsHelpful() work? I know that it is pseudocode, but the paper does not give sufficient detail to implement the algorithm.
- How is 'brainish' different from natural language? If I understand it correctly, in the CMT, brainish is supposed to be inherently multimodal, not explicitly symbolic or language like. Can gists contain anything other than language, if so, how can all models understand them (if there's an image in a gist, an ALM won't be able to process it).
- What was process like of translating the original CTM to the LM-based CTM-AI? How did the authors decide which aspects to include? For example, in the original CTM, the 'Model of the World' LTM plays quite a crucial role, yet it does not appear in the CTM-AI. Did the authors not deem it important? Or did they try it and it didn't work?
- More generally, what is the author's perspective on how the CMT-AI relates to the CMT framework? Was the CMT simply an inspiration? Does the fact that CMT-AI works support the validity of CMT as a model of consciousness? Would this then mean that CMT-AI is conscious? I think dealing with these concepts, even if they just appear in the name, requires a careful discussion of such questions.
- The title overpromises; it is very unclear to me to what extent the proposed framework is a blueprint for general AI, and also how exactly it inspired by consciousness (as opposed to simply by some aspects of the CMT model).

Some additional small comments and corrections:

- P.8: Figure 3 is mentioned twice, but it doesn't exist. I assume the authors mean Table 3?
- P.9 line 467 "and explicitly asking for more surrounding context" is grammatically incorrect in this context.
- Appendix E: Please reference Table 7, it might not be immediately obvious that the table belongs to the section. A more detailed analysis could be insightful.

---

> ### Author Response · Authors · 2025-11-21
>
> # Experimental details
> We include multi-pages additional details in the revised version of our paper in the `Appendix C` covered in blue font.
>
> **Hyper-parameters**: We include the hyper-parameter settings of our model inference in `Appendix C.1`.
>
> **Baseline details**: We include details of the baseline models (explanation, citation, model params, fine-tuned or prompting-based models) in `Appendix C.2`.
>
> # CTM-AI implementation details
> In the revised paper, we added multi-pages of substantial clarification in `Appendix D` (highlighted in blue), directly addressing all reviewer’s questions:
>
> **Processor settings:** `Appendix D.1` and `Appendix D.2` list the number and configuration of processors for all four benchmarks.
>
> **Processor prompting:** `Appendix D.1` (STM) and `Appendix D.3` (LTM) provide the exact prompts used. These prompts mainly define the task and modality information and are shared with the base-model baselines; no task-specific prompt engineering is required. We only tried a few times for collecting reasonable improvement on benchmarks.
>
> **Weight collection:** `Appendix D.3` explains how weights are computed using relevance, confidence, and surprise with a 1:1:0.2 ratio.
>
> **Link formation:** `Appendix D.6` describes how processor links are formed, including prompts and thresholds.
>
> **Pseudo code:** `Appendix D.8` provides streamlined, function-free pseudocode covering all stages of CTM-AI inference, and Figure 4 visualizes the overall dynamics.
>
>
>
>
>
>
>
> # Cost and efficiency analysis
>
>
> **Algorithmic analysis** `Appendix D.8` provides a theoretical characterization of CTM-AI’s cost. Because processors run in parallel, the expected runtime is about 3 times that of a single base model. Token usage per example follows $2 \times \text{processors} + 2 \times \text{links}$, reflecting the parallel competition and global-broadcast steps. Potential optimization involves ensuring the sparsity of the processor graph and parallel multi-query inference for each processor.
>
> **Empirical analysis** We additionally report empirical measurements including `token cost`, `number of API calls`, and `wall-clock latency` per data point. We ensure fair comparison by matching CTM-AI with multi-agent baselines using similar token budgets.
>
>
> (1) **multi-agent debate**: multiple agents adopt different stances and debate sequentially, with a judge synthesizing the final answer; this process cannot be parallelized.
>
> (2) **multi-agent orchestra**: a controller decomposes the sarcasm task into sub-queries, an executor answers them, and a summarizer aggregates the results; only the sub-query execution stage is parallelizable.
>
>
> CTM-AI’s average inference time is ~3× that of the base model, since many examples terminate early after the first competition stage. The average number of API calls per example is ~8, which is consistent with the theoretical token-cost estimate. The performance is better than similar cost baselines.
>
> | MUStARD               | Token Cost per DP | API call per DP | Latency per DP | F1     |
> |-----------------------|-------------------|------------------|-------------|--------|
> | single-agent (base model)          | $0.00022          | 1                | 2.62s       | 0.5072 |
> | multi-agent debate    | $0.0028           | 11               | 27.67s      | 0.6423 |
> | multi-agent orchestra | $0.0019           | 9                | 7.04s      | 0.6555 |
> | CTM-AI                | $0.0014           | 8.22             | 6.47s       | 0.7377 |
>
>
>
>
> | URFunny | API call per DP | F1 |
> | -------- | -------- | -------- |
> | single-agent (base model)     | 1     | 60.66     |
> | CTM-AI     | 7.92     | 71.62     |
>
>
> # Comparison with NLSOMs (Zhuge et al.)
> We point out two key differences and our contribution compared with Zhuge et al's NLSOMs:
>
> **Role Assignment vs. Ability-Driven Interaction**.
> NLSOMs rely on predefined roles (e.g., leader, organizer, critic), keeping them within the traditional multi-agent orchestration paradigm. CTM-AI does not assign roles; interactions emerge naturally from processors’ heterogeneous abilities (e.g., vision processors querying text processors), rather than from scripted coordination patterns.
>
> **Cognitive Foundations**.
> CTM-AI is directly grounded in CTM consciousness theory and follows its architectural principles, which emphasize parallel sensing and general-purpose information integration. This cognitively motivated design enables broad generalization across diverse tasks. NLSOMs, by contrast, are not based on a concrete consciousness theory but the ambigous concept of society of minds and rely more heavily on engineered coordination workflows.

---

> ### Author Response · Authors · 2025-11-21
>
> # Relationship with Branish
>
> The Branish language in [1] is designed for multimodal **fusion, translation, and generation**. Because Branish is difficult to formalize for practical use, we approximate it with English. All processors output their gists in English, and modern multimodal LLMs can reliably interpret and generate English. This allows English to fulfill Branish’s three roles—**fusion** (e.g., combining vision and text descriptions in English), **translation** (e.g., expressing visual features in text), and **generation** (e.g., producing a final answer in English)—making it a practical substitute for Branish in CTM-AI.
>
> # Relationship with CTM theory
>
> **Connection with CTM Theory** We implement all core mechanisms from CTM theory—LTM/STM processors, up-tree and down-tree broadcasting, link formation, and multimodal fusion—using modern AI techniques such as in-context learning, multimodal foundation models, LLM-as-a-judge prompting, and function calling. Down-tree broadcasting is realized through memory updates for in-context learning, while processor links and multimodal fusion are implemented via mutual querying in text space. In this sense, CTM-AI serves as a practical instantiation of the CTM architecture, with empirical approximations tailored to current LLM technologies.
>
> **Processor selection** As an ML-oriented paper rather than a purely cognitive one, we must produce benchmark results under limited computational resources. Since we cannot afford a system with thousands of processors, we heuristically select the most relevant modality- or function-specific processors for each benchmark. The notion of a “model of the world” in CTM theory is broader and more ambiguous; current implementations of "world model" often refer to video generation models, which we do not evaluate because our tasks do not require such capabilities.
>
> # Relationship with consciousness
> CTM-AI is inspired by the Conscious Turing Machine (CTM), a theoretical framework about consciousness. We do not claim that CTM-AI is conscious; rather, CTM serves as a principled blueprint for a practical architecture. A system grounded in a consciousness theory is expected to support general-purpose cognitive functions, motivating our focus on generality. CTM-AI can flexibly integrate diverse processors—vision, language, audio, tools—and operate across a broad range of task domains.
>
> # Overpromise title
> We would change the title of our paper to be a prototype framework inspired by a consciousness theory (CTM) without emphasizing general AI in the later version.
>
> # Typo and grammar errors
> We have updated all the mentioned typos and grammatical errors in our revised version of the paper. Please check the part marked in blue in the main content.
>
> We hope these discussions are helpful for answering the valuable questions raised by reviewers. If the concerns are resolved, we kindly request the reviewer to consider increasing the score. We are happy to discuss more if there is any further questions.
>
> [1] Liang et al. Brainish: Formalizing A Multimodal Language for Intelligence and Consciousness

---

> > ### Author Response · Authors · 2025-11-26
> >
> > Dear Reviewer,
> >
> > As we approach the end of the discussion phase, please let us know if there are any remaining questions or points we can help clarify. We truly appreciate your feedback and would be glad to provide any additional information that could support your assessment.
> >
> > Bests,
> >
> > Authors

---

### Official Review · Reviewer_JtxT · 2025-11-01

**Soundness:** 4
**Presentation:** 3
**Contribution:** 3
**Rating:** 4
**Confidence:** 3

**Summary:**

This paper presents blueprint for implementing a general AI system based on the Conscious Turing Machine (CTM), a formal machine model of consciousness. Unlike other cognitive architectures, CTM employs a global workspace and distributed competition and collects parallel independent cognitive, sensory, motor, and extended processors.

**Strengths:**

Strengths
- Practical application of CTM model by CTM-AI
- Empirical demonstration of CTM-AI as a general and multi-action AI, including language modeling, multimodal understanding and human behavior understanding

**Weaknesses:**

- The proposed algorithm is dependent on finite lifetime T and height h of binary tree, Up-Tree. Can the author provide an empirical evaluation of how these parameters impact model’s performance? An example of such analysis could the the answer of RQ1 of the following paper [1]

-Can the author compare their proposed methods alternative approaches (e.g., ensemble voting, query augmentation, or multi-agent debates?)



References
Saha, Swarnadeep, Peter Hase, and Mohit Bansal. "Can language models teach? teacher explanations improve student performance via personalization." Advances in Neural Information Processing Systems 36 (2023): 62869-62891.

**Questions:**

Please see weakness.

---

> ### Author Response · Authors · 2025-11-21
>
> # Analysis on lifetime T
> In CTM theory, the lifetime $T$ explains the slow up-tree and fast down-tree processes: with millions of processors, centralized competition becomes memory-intensive, so CTM uses local up-tree competition to reduce comparisons, while down-tree broadcasting remains fast. In our implementation, $T$ for chunks is related to the number of inference iterations. Ablations on MUsTARD show that increasing the iteration count improves performance in sarcasm detection.
>
>
>
> | MUsTARD | iteration=1 |  adaptive iteration (ours avg iteration: 2.26)   | iteration=3 |
> | -------- | -------- | --- | -------- |
> | CTM-AI (F1)     | 68.99%     |  73.77%   | 82.5%     |
>
>
> # Analysis on binary tree height h
> CTM theory uses a binary tree for up-tree sampling (Line 142–144), but in our setting the number of processors is small, so we use direct global competition; effectively, the tree height is fixed to 1. Because CTM’s up-tree mechanism ensures location-independent winning probabilities, deeper trees with more processors would behave similarly to our global competition—the only difference would be memory constraints at large scale.
>
> # Relation to referenced paper
> The referenced paper’s RQ1 asks: “Can a teacher LLM intervene at test time to improve a student LLM’s predictions?”
> In contrast, our framework does not rely on a teacher–student setup. Instead, there is no privileged teacher model—all processors operate symmetrically.
>
> However, our **[Analysis on timespan T]** shows that increasing the number of iterative steps with a fixed processor set naturally strengthens multimodal fusion and allows the system to **self-correct** over iterations. In this sense, CTM-AI provides a mechanism-guaranteed, test-time intervention that improves prediction quality without requiring an external teacher.
>
> # Additional baseline
> We evaluate multi-agent debate, query augmentation, and ensemble voting under comparable or higher token costs. CTM-AI averages 8.22 processor calls on MUStARD and 7.98 on URFUNNY, yet consistently outperforms these baselines.
>
> (1) **Multi-agent debate**: agents argue from different viewpoints and a judge decides the final answer; debate depth is capped at 11 API calls per example.
>
> (2) **Query augmentation**: a controller creates 7 sub-queries, an executor answers them, and a summarizer combines the results, averaging 9 API calls.
>
> (3) **Ensemble voting**: the same 3 processors as CTM-AI each run 4 times, totaling 12 API calls per example.
>
>
>
> | Mustard       | Single model | Multi-agent debate | Query augmentation | Ensemble voting (T=0.9) | CTM-AI |
> | ------------- | ------------ | ------------------ | ------------------------------ | ------------------------------ | ------ |
> | **#API call**| 1 | 11 | 9 | 12 | 8.22|
> | **Acc**       | 0.5900       | 0.5149             | 0.6239                         | 0.6190                         | **0.7388** |
> | **F1**        | 0.5072       | 0.6423             | 0.6555                         | 0.6444                         | **0.7377** |
> | **Recall**    | 0.5900       | 0.8627             | 0.6610                         | 0.6444                         | **0.7444** |
> | **Precision** | 0.7747       | 0.5116             | 0.6500                         | 0.6444                         | **0.7396** |
>
>
> | URFunny        | Baseline | Ensemble Voting (T=0.9) | CTM-AI    |
> | ------------- | -------- | ---------------------- | ------ |
> | **#API call**| 1 | 12 | 7.98|
> | **Acc**       | 0.5200   | 0.5510                 | 0.6869 |
> | **F1**        | 0.5198   | 0.5224                 | 0.6822 |
> | **Recall**    | 0.5200   | 0.5510                 | 0.6882 |
> | **Precision** | 0.5200   | 0.5671                 | 0.7010 |
>
>
> We hope these discussions are helpful for answering the valuable questions raised by reviewers. If the concerns are resolved, we kindly request the reviewer to consider increasing the score. We are happy to discuss more if there is any further questions.

---

> > ### Author Response · Authors · 2025-11-26
> >
> > Dear Reviewer,
> >
> > As we approach the end of the discussion phase, please let us know if there are any remaining questions or points we can help clarify. We truly appreciate your feedback and would be glad to provide any additional information that could support your assessment.
> >
> > Bests,
> >
> > Authors

---

### Official Review · Reviewer_dnVs · 2025-11-01

**Soundness:** 3
**Presentation:** 3
**Contribution:** 3
**Rating:** 6
**Confidence:** 3

**Summary:**

This paper proposes CTM-AI, a multi-processor architecture inspired by the Conscious Turing Machine. Specialized and general processors compete to write into a global workspace (STM), whose content is then broadcast; processors can also form dynamic links. The system is evaluated on multimodal perception, tool use, and small web-agent tasks and shows improvements over reported baselines. Ablation studies convincingly validate each component. However, baseline comparisons all use 2021-2023 models (vs. 2024 Gemini-2.0),  and offers limited proof of cross-modal balancing. These are addressable issues.

**Strengths:**

1. Clear presentation with solid ablation studies validating each component's contribution.

2. The closed-loop mechanism of module competition, STM storage, and global broadcasting is well-motivated and modular, facilitating transfer to other tasks.
3. Demonstrates results on multimodal understanding, tool use, and small web-agent tasks, suggesting the approach is not task-specific.

**Weaknesses:**

1. CTM-AI uses Gemini-2.0-flash-lite (2024) while baselines are from 2021-2023 (MMoE, BLIP2, etc.), making it difficult to isolate architectural contributions from model improvements.

2. Efficiency reporting (suggested). It would strengthen the paper to include average end-to-end latency, API calls per task, and cost relative to the base model to contextualize practical overheads.

**Questions:**

1. How are rare or weak modalities weighted or gated so they are not overshadowed by dominant, feature-rich modalities (e.g., vision/text)?

2. Could you share systematic failure modes (e.g., vision-only misleads, unhandled tool errors) and the mitigations you found effective?

---

> ### Author Response · Authors · 2025-11-21
>
> # Baseline discussion
> We compare CTM-AI against two types of baselines: (1) **prompting baselines** such as Gemini-2.0-flash-lite (base model), which also serves as the base processor in CTM-AI; and (2) **state-of-the-art fine-tuned models** for each dataset, such as MMoE models fine-tuned on URFUNNY and MUStARD. `Appendix C`  (line 771-814) sin the revised paper provides full details on model parameters, training or prompting, and baseline configurations.
>
>
> # Cost and efficiency analysis
>
> **Algorithmic analysis** `Appendix D.8` provides a theoretical characterization of CTM-AI’s cost. Because processors run in parallel, the expected runtime is about 3 times that of a single base model. Token usage per example follows $2 \times \text{processors} + 2 \times \text{links}$, reflecting the parallel competition and global-broadcast steps. Potential optimization involves ensuring the sparsity of the processor graph and parallel multi-query inference for each processor.
>
> **Empirical analysis** We additionally report empirical measurements including `token cost`, `number of API calls`, and `wall-clock latency` per data point. We ensure fair comparison by matching CTM-AI with multi-agent baselines using similar token budgets.
>
>
> (1) **multi-agent debate**: multiple agents adopt different stances and debate sequentially, with a judge synthesizing the final answer; this process cannot be parallelized.
>
> (2) **multi-agent orchestra**: a controller decomposes the sarcasm task into sub-queries, an executor answers them, and a summarizer aggregates the results; only the sub-query execution stage is parallelizable.
>
>
> CTM-AI’s average inference time is ~3× that of the base model, since many examples terminate early after the first competition stage. The average number of API calls per example is ~8, which is consistent with the theoretical token-cost estimate. The performance is better than similar cost baselines.
>
> | MUStARD               | Token Cost per DP | API call per DP | Latency per DP | F1     |
> |-----------------------|-------------------|------------------|-------------|--------|
> | single-agent (base model)          | $0.00022          | 1                | 2.62s       | 0.5072 |
> | multi-agent debate    | $0.0028           | 11               | 27.67s      | 0.6423 |
> | multi-agent orchestra | $0.0019           | 9                | 7.04s      | 0.6555 |
> | CTM-AI                | $0.0014           | 8.22             | 6.47s       | 0.7377 |
>
>
>
>
> | URFunny | API call per DP | F1 |
> | -------- | -------- | -------- |
> | single-agent (base model)     | 1     | 60.66     |
> | CTM-AI     | 7.92     | 71.62     |
>
>
> # Modality weighting
> Processors specializing in different modalities can win either by directly contributing useful information or by synergizing with other modalities. Thus, the multimodal fusion mechanism ensures that no single modality is overshadowed. Empirically, on MUStARD, the three modality-specific processors exhibit similar win rates, consistent with the fact that sarcasm detection requires balanced multimodal cues.
>
> | Processor win rate | Vision |  Audio   | Text | Avg Iteration Num
> | -------- | -------- | --- | -------- | -------- |
> | CTM-AI     | 39.83%     |  38.14%   | 22.03%     | 2.24 |
>
>
>
> # Failure modes
> In the revised paper, we provide additional case studies. `Figure 5` (Page 21) illustrates single-modality misleading cases, and `Figure 6` (Page 22) shows tool-use misleading cases. We find that a simple remedy—using more iterations and a higher output-confidence threshold—effectively mitigates both failure modes.
>
>
> | MUStARD | iteration=1 |  adaptive iteration (ours avg iteration: 2.26)   | iteration=3 |
> | -------- | -------- | --- | -------- |
> | CTM-AI (F1)     | 68.99%     |  73.77%   | 82.5%     |
>
> We hope these discussions are helpful for answering the valuable questions raised by reviewers. If the concerns are resolved, we kindly request the reviewer to consider increasing the score. We are happy to discuss more if there is any further questions.

---

> > ### Author Response · Authors · 2025-11-26
> >
> > Dear Reviewer,
> >
> > As we approach the end of the discussion phase, please let us know if there are any remaining questions or points we can help clarify. We truly appreciate your feedback and would be glad to provide any additional information that could support your assessment.
> >
> > Bests,
> >
> > Authors

---

### Official Review · Reviewer_Wsj6 · 2025-11-01

**Soundness:** 2
**Presentation:** 2
**Contribution:** 2
**Rating:** 4
**Confidence:** 5

**Summary:**

This paper proposes a hierarchical method for enhancing integration of multi-modal perceptions for abstract decision making and tool using. This applies the mechanism of the “Conscious Turing Machine” to current multimodal pre-trained large models. The experiments somehow show improvements on classification tasks, long term decision tasks, and tools using tasks with multi-modal models without reinforcement post-raining. However, some concern arises about the implementation methods, and discussions about the reliability are missing for some of the concepts defined. And the reinforcement learning models are not compared in terms of both theoretical and empirical justification of absolute improvement orthogonal to the advantages introduced by post-training, which makes the contributions hard to identify.

**Strengths:**

1. The topic discussed is new, which tries to build an agentic framework that is biomimetic.
2. The paper is easy to follow in terms of what the authors intend to show.
3. The authors try hard to provide empirical evidence that the framework has superiority over “monolithic” models such as GPT-4o and Gemini-Flash-Lite.

**Weaknesses:**

1. The generation procedure of “utility weight”, defined as an element of the chunks, is not introduced. Moreover, the definition as “certainty or utility” makes it hard to buy it.
2. Authors should add more discussion to justify in what circumstances the advantages can be achieved compared to other advanced techniques, such as post-training. To achieve this, rigorous theoretical discussions or a more serious demonstration of the ablation study should be provided.
3. The efficiency of the provided method and how to optimize it are not provided in the manuscript.
4. The computation consumption & parameter number of the baselines is not clearly introduced and compared in the manuscript, which leads to fairness concerns.

**Questions:**

See weaknesses.
Further question as following:
How the proposed method processes nested complex structure in each perception that is chunked to a swollen chunk?

---

> ### Author Response · Authors · 2025-11-21
>
> # Chunk generation details
>
> **Chunk weight generation.** Each chunk’s weight is computed as a combination of *relatedness*, *confidence*, and *surprise*, following the definitions in the CTM theory paper. These metrics respectively capture whether the processor believes the gist is relevant to the query, obvious, and informative beyond expectation. The final weight score is a weighted combination of the three. `Appendix D.3` (line 872-902) in the revised paper provides detailed formulas and prompts.
>
> **Chunk nested-structure generation.** We obtain the hierarchical chunk structure directly via prompting by asking the model to output a JSON-formatted nested representation. `Appendix D.3` (line 872-902) aincludes the full prompt and examples used in this procedure.
>
> # Comparison with post-training
>
> **Intuitive analysis** CTM-AI is most advantageous when multimodal interactions are complex, interdependent, and require multi-round cross-modal reasoning—such as multi-tool use or affective computing. These tasks demand flexible coordination across heterogeneous modalities (vision, audio, text, tools) that cannot be easily captured by a single centralized representation. Standard post-training struggles here because it relies on large amounts of tightly aligned multimodal data, which is extremely scarce [1]. As a result, it often fails to learn the rich cross-modal communication patterns needed for such scenarios.
>
>
> **Theoretical analysis** From a theoretical perspective, centralized multimodal post-training implicitly assumes that a *single* parameter vector $\theta$ can lie near a Pareto-optimal point of the multi-objective performance vector:
>
> $\mathbf{J}(\theta) = (J_1(\theta), \dots, J_M(\theta))$
>
> where each $J_m$ represents a different modality or tool. Standard post-training enforces this by optimizing a scalarized loss $\sum_m \lambda_m \mathcal{L}_m(\theta)$, forcing all modalities to share one representation and one trade-off point on the Pareto front—an assumption that is fragile when synergistic multimodal data is scarce[2, 3].
>
> CTM-AI relaxes this assumption not by training, but by keeping modality-specific processors fixed and optimizing the communication dynamics through which they exchange and refine information.Formally, instead of searching for a single $\theta$, CTM-AI operates over a coordination objective：
>
> $\max_{\text{comm., dynamics}} \Phi\big(J_1, \dots, J_P\big)$,
>
> where $\Phi$ is induced by the up/down-tree mechanism that integrates partial signals from different processors. This shifts multimodal learning from aligning all modalities into one representation to optimizing their interaction, enabling more robust cross-modal reasoning.
>
>
> # Ablation study
>
> We provide detailed ablations for each inference component of CTM-AI in `Appendix D` (line 815-957). Because CTM-AI is purely an inference algorithm and involves no parameter training, we already include ablations on all design choices. Moreover, we compare CTM-AI against strong multi-agent inference baselines—including multi-agent orchestra, multi-agent debate, and ensemble voting—to evaluate its effectiveness under comparable settings.
>
> | Mustard       | Single model | Multi-agent debate | Multi-agent Orchestra | Ensemble voting | CTM-AI |
> | ------------- | ------------ | ------------------ | ------------------------------ | ------------------------------ | ------ |
> | #API call per datapoint | 1 | 11 | 9 | 12 | 8.22|
> | Acc       | 0.5900       | 0.5149             | 0.6239                         | 0.6190                         | **0.7388** |
> | F1        | 0.5072       | 0.6423             | 0.6555                         | 0.6444                         | **0.7377** |
> | Recall    | 0.5900       | 0.8627             | 0.6610                         | 0.6444                         | **0.7444** |
> | Precision | 0.7747       | 0.5116             | 0.6500                         | 0.6444                         | **0.7396** |

---

> ### Author Response · Authors · 2025-11-21
>
> # Cost and efficiency analysis
>
> **Algorithmic analysis** `Appendix D.8` (line 941-957) provides a theoretical characterization of CTM-AI’s cost. Because processors run in parallel, the expected runtime is about 3 times that of a single base model. Token usage per example follows $2 \times \text{processors} + 2 \times \text{links}$, reflecting the parallel competition and global-broadcast steps. Potential optimization involves ensuring the sparsity of the processor graph and parallel multi-query inference for each processor.
>
> **Empirical analysis** We additionally report empirical measurements including `token cost`, `number of API calls`, and `wall-clock latency` per data point. We ensure fair comparison by matching CTM-AI with multi-agent baselines using similar token budgets.
>
>
> (1) **multi-agent debate**: multiple agents adopt different stances and debate sequentially, with a judge synthesizing the final answer; this process cannot be parallelized.
>
> (2) **multi-agent orchestra**: a controller decomposes the sarcasm task into sub-queries, an executor answers them, and a summarizer aggregates the results; only the sub-query execution stage is parallelizable.
>
>
> CTM-AI’s average inference time is ~3× that of the base model, since many examples terminate early after the first competition stage. The average number of API calls per example is ~8, which is consistent with the theoretical token-cost estimate. The performance is better than similar cost baselines.
>
> | MUStARD               | Token Cost per DP | API call per DP | Latency per DP | F1     |
> |-----------------------|-------------------|------------------|-------------|--------|
> | single-agent (base model)          | $0.00022          | 1                | 2.62s       | 0.5072 |
> | multi-agent debate    | $0.0028           | 11               | 27.67s      | 0.6423 |
> | multi-agent orchestra | $0.0019           | 9                | 7.04s      | 0.6555 |
> | CTM-AI                | $0.0014           | 8.22             | 6.47s       | 0.7377 |
>
>
>
>
> | URFunny | API call per DP | F1 |
> | -------- | -------- | -------- |
> | single-agent (base model)     | 1     | 60.66     |
> | CTM-AI     | 7.92     | 71.62     |
>
>
> # Baseline discussion
> We compare CTM-AI against two types of baselines: (1) **prompting baselines** such as Gemini-2.0-flash-lite (base model), which also serves as the base processor in CTM-AI; and (2) **state-of-the-art fine-tuned models** for each dataset, such as MMoE models fine-tuned on URFUNNY and MUStARD. `Appendix C`  (line 771-814) in the revised paper provides full details on model parameters, training or prompting, and baseline configurations.
>
> We hope these discussions are helpful for answering the valuable questions raised by reviewers. If the concerns are resolved, we kindly request the reviewer to consider increasing the score. We are happy to discuss more if there is any further questions.
>
> [1] Yu et al. MMoE: Enhancing Multimodal Models with Mixtures of Multimodal Interaction Experts
>
> [2] Navon et al. Learning the Pareto Front with Hypernetworks
>
> [3] Yan et al. A MultiModal Targeted Pareto Framework for Fake News Detection

---

> > ### Author Response · Authors · 2025-11-26
> >
> > Dear Reviewer,
> >
> > As we approach the end of the discussion phase, please let us know if there are any remaining questions or points we can help clarify. We truly appreciate your feedback and would be glad to provide any additional information that could support your assessment.
> >
> > Bests,
> >
> > Authors

---

### Official Review · Reviewer_i9GU · 2025-11-01

**Soundness:** 3
**Presentation:** 4
**Contribution:** 3
**Rating:** 6
**Confidence:** 4

**Summary:**

This paper presents CTM-AI, a practical framework that operationalizes the theoretical Conscious Turing Machine (CTM) model. The architecture is designed to mimic principles of consciousness by using a large number of distributed processors that operate in parallel. These processors are not limited to modalities but include specialized experts for sensory input, cognitive reasoning, and extended tool/API use. Instead of a central executive, the system uses a "global workspace" (Short-Term Memory) where processors compete via an "up-tree" mechanism to select a winning "chunk" of information. This winning chunk is then broadcast via a "down-tree" to all other processors, enabling dynamic information fusion and iterative reasoning.

**Strengths:**

S1) The core contribution of bridging the gap between the abstract Conscious Turing Machine (CTM) theory and a concrete, implementable AI system is novel and interesting6.

S2) The paper provides a clear explanation of its dynamics, supported by strong ablation studies (Table 3, 5) and an intuitive case study (Figure 2) that effectively demonstrates the iterative reasoning process.

S3) The experimental evaluation is rigorous, testing the framework's versatility across a diverse set of tasks, including multimodal perception (MUSTARD), tool use (StableToolBench), and agentic web tasks (WebArena).

**Weaknesses:**

O1)  Distinction with multi-agent systems: The paper's primary claim of differentiation from multi-agent systems rests on CTM-AI being "free of a central executive" and not using "fixed role assignments". However, this distinction feels underspecified. The "Up-Tree competition" mechanism, which selects the chunk with the highest weight, and the "STM processor" (a stateless LLM that produces the final answer)  functionally serve as a form of selection and orchestration. The paper would be stronger if it more directly contrasted its dynamic, competition-based mechanism against a modern multi-agent orchestrator on a complex task.

O2) Discrepancy Between Theory and Implementation: The paper introduces the formal CTM with an "enormous number of powerful processors". It then abandons the CTM's core hierarchical, local competition mechanism in favor of a simple "global competition" (argmax). The justification is that "typically only a few (<10) LTM processors are active". This simplification, motivated by a small-scale implementation, undermines the scalability and theoretical grounding the CTM framework is supposed to provide.

O3 ) Ambiguous Learning Mechanism: The paper's definition of "learning" is weak and underdeveloped. Section 3.3 describes learning as "in-context learning with memory updates", which consists of broadcasting the winning chunk to all processors' memories and updating the link matrix. It is unclear if the processor parameters (theta_i) themselves are ever updated. If all "learning" is purely modifying the LTM (prompt memory) and processor graph, this is a significant limitation and may not support the deep adaptation or skill acquisition implied by the "general AI" goal.

**Questions:**

Please address weaknesses O1-O3.

---

> ### Author Response · Authors · 2025-11-21
>
> # Differences with multi-agent system
> **Conceptual comparison** We compare multi-agent frameworks like AgentVerse[1] and our proposed CTM-AI in four dimensions.
>
> | Dimension | Multi-agent System | CTM-AI |
> |----------|---------------------|--------|
> | Agent Roles | Agents follow predefined, prompt-specified roles. | Processors have distinct **capabilities** (vision, audio, tools) and independently interpret the same multimodal input. |
> | Control Mechanism | A central controller orchestrates planning and assigns sub-tasks to agents. | The STM-processor provides global feedback/output formatting; **no task routing**—feedback is broadcast to all processors. |
> | Workflow Pattern | Agents operate **sequentially**, creating dependency chains and potential error propagation. | Processors operate **in parallel**, reflecting cognitive/global-workspace dynamics. Inter-processor dependencies emerge naturally via processor links. |
> | Use Case | Effective for math, coding, and structured tasks requiring explicit procedural decomposition. | Suited for multisensory and cognitively complex tasks (e.g., sarcasm, humor) while still supporting problem-solving tasks without manual pipelines. |
>
>
> **Empirical comparison** We further include strong multi-agent baselines to compare performance under matched token costs.
>
> (1) **Multi-agent debate**: agents adopt different opinions and debate sequentially, with a judge producing the final answer.
>
> (2) **Multi-agent orchestra**: a controller decomposes the query into sub-queries, an executor answers them from multiple perspectives, and a summarizer aggregates the results.
>
> (3) **Ensemble voting**: uses the same 3 processors as CTM-AI but simply applies 4 times of majority voting over multiple independent inferences.
>
> On average, CTM-AI invokes 8.22 processor calls per MUStARD example and 7.98 per URFUNNY example. Despite operating at similar token/processor cost, CTM-AI consistently outperforms all multi-agent baselines, demonstrating that its performance gains come from the cognitive architecture rather than increased computation.
>
> | Mustard       | Single model | Multi-agent debate | Multi-agent Orchestra | Ensemble voting | CTM-AI |
> | ------------- | ------------ | ------------------ | ------------------------------ | ------------------------------ | ------ |
> | #API call per datapoint | 1 | 11 | 9 | 12 | 8.22|
> | Acc       | 0.5900       | 0.5149             | 0.6239                         | 0.6190                         | **0.7388** |
> | F1        | 0.5072       | 0.6423             | 0.6555                         | 0.6444                         | **0.7377** |
> | Recall    | 0.5900       | 0.8627             | 0.6610                         | 0.6444                         | **0.7444** |
> | Precision | 0.7747       | 0.5116             | 0.6500                         | 0.6444                         | **0.7396** |
>
>
> | URFunny        | Baseline | Ensemble Voting | CTM-AI    |
> | ------------- | -------- | ---------------------- | ------ |
> | #API call per datapoint | 1 | 12 | 7.98|
> | Acc       | 0.5200   | 0.5510                 | 0.6869 |
> | F1        | 0.5198   | 0.5224                 | 0.6822 |
> | Recall    | 0.5200   | 0.5510                 | 0.6882 |
> | Precision | 0.5200   | 0.5671                 | 0.7010 |
>
>
>
>
> # Up-tree competition mechanism in CTM-AI
> In the original CTM theory, designed local competition ensures location-independent selection while reducing memory cost during up-tree competition. Because our setting uses only a small number of LTM processors and faces no memory constraints, we simplify this to a global competition across processors, which also scales naturally when sampling over larger processor sets.
>
> CTM defines sampling with an additional competition function
>
> $f(\text{chunk}) = c\big(\text{intensity} + d(\text{mood})\big)$,
>
> However, VLM/LLM processors already introduce sufficient randomness through their generation process. We therefore adopt a deterministic **argmax** competition, which is empirically more stable and corresponds to taking $c \to \infty$ in the CTM formulation—remaining fully consistent and mathematically grounded with CTM theory.
>
>
> # Learning mechanism in CTM-AI
> CTM-AI currently updates only processor memories and the processor-graph structure while leaving model parameters fixed. This is reasonable because in-context learning already provides continual-learning capabilities comparable to fine-tuning in large foundation models [2,3]. Moreover, recent online RL methods show that foundation models can effectively use verbal feedback for parameter updates [4]. Building on this, future versions of CTM-AI can treat the feedback exchanged during up-tree competition as verbal feedback, enabling RL-based parameter updates within the same framework.

---

> > ### Author Response · Authors · 2025-11-21
> >
> > # Scalability of CTM-AI
> > To show scalability, we tested CTM-AI with **20 processors** on StableToolBench by randomly sampling 20 tools per query (including all required ones). Across five runs, performance matched the setting where only relevant processors were used. Irrelevant processors receive low relevance scores, fail to generate useful responses, lose in the up-tree competition, and form no links—thus having no impact on the final output. This confirms that CTM-AI scales robustly even with many extraneous processors.
> >
> >
> > We hope these discussions are helpful for answering the valuable questions raised by reviewers. If the concerns are resolved, we kindly request the reviewer to consider increasing the score. We are happy to discuss more if there is any further questions.
> >
> > [1] Chen et al. AgentVerse: Facilitating Multi-Agent Collaboration and Exploring Emergent Behaviors
> >
> > [2] Kang et al. In-Context Learning can Perform Continual Learning Like Humans
> >
> > [3] Mosbach et al. Few-shot Fine-tuning vs. In-context Learning: A Fair Comparison and Evaluation
> >
> > [4] Luo et al. Language Models Can Learn from Verbal Feedback Without Scalar Rewards

---

> > > ### Author Response · Authors · 2025-11-26
> > >
> > > Dear Reviewer,
> > >
> > > As we approach the end of the discussion phase, please let us know if there are any remaining questions or points we can help clarify. We truly appreciate your feedback and would be glad to provide any additional information that could support your assessment.
> > >
> > > Bests,
> > >
> > > Authors

---

### Author Response · Authors · 2025-11-21
**Summary of Rebuttal: Consensus on Strengths & Resolution of Concerns**

# General Response

We appreciate the reviewers’ efforts in evaluating our paper. Below, we summarize the key points reviewers raised—items marked with ** indicate issues for which we provide additional experiments or clarifications, while unmarked items reflect strengths acknowledged by them. "Action/Summary" includes the highly summarized rebuttal content for each reviewer.


|                     | Reviewer i9GU | Reviewer Wsj6 | Reviewer dnVs | Reviewer JtxT | Reviewer NwUJ | Action/Summary |
| ------------------- | ------------- | ------------- | ------------- | ------------- | ------------- | -------------- |
| **Novelty** | "...implementable AI system is novel and interesting" | "The topic discussed is new..." | "...is well-motivated and modular,..." | NA | "...opens up a fascinating perspective..." | `summary`: 4 out of 5 reviewers think the paper is novel and interesting. |
| **Presentation** | "...provides a clear explanation of its dynamics..." | "...The paper is easy to follow" | "Clear presentation with..." | NA | \*\***"lack of detail in the implementation and experimental protocol..."** | `summary`: 3 out of 5 reviewers think this paper is clearly presented. `Reviewer NwUJ rebuttal`: We add full implementation and experimental details in the Appendix of the revised version. |
| **Evaluation** | "...testing...across a diverse set of tasks" | "try hard to provide empirical evidence" | "Demonstrates results on...the approach is not task-specific" | "Empirical demonstration of CTM-AI as a general and multi-action AI..." | "CTM-AI performs competitively with...several baselines" | `summary`: All reviewers believe this paper is evaluated on a diverse range of tasks and achieves relatively good performance. |
| **Baseline** | \*\***"...be stronger if it more directly contrasted its dynamic, competition-based mechanism against a modern multi-agent orchestrator..."** | \*\***"...computation consumption & parameter number of the baselines is not clearly introduced and compared"** | \*\***"CTM-AI uses Gemini(2024) while baselines are from 2021-2023 (MMoE, BLIP2, etc.), making it difficult to isolate..."** | \*\***"...should compare with alternative approaches (e.g., ensemble voting, query augmentation, or multi-agent debates?)"** | NA | `Reviewer i9GU & JtxT rebuttal`: We add multi-agent orchestrator, ensemble voting, query augmentation, and multi-agent debate baselines with similar cost; all perform worse than CTM.`Reviewer Wsj6 & dnVs rebuttal`: We add baseline details in the Appendix. |
| **Cost/Efficiency Discussion** | NA | \*\***"The efficiency of the provided method and how to optimize..."** | \*\***"Efficiency reporting (suggested)..."** | NA | \*\***"...How does token/FLOP compare to the baselines?"** | `Reviewer Wsj6 & dnVs & NwUJ rebuttal`: We emphasize efficiency is *not* the central focus of our cognitive-driven prototype. We add theoretical + empirical analysis of per-data-point **API call count**, **token cost**, and **wall-clock latency**, making fair comparison between CTM-AI and multi-agent system/ensembling baselines with similar inference costs. |
| **Relationship with CTM Theory** | \*\***"...abandons the CTM's core hierarchical, local competition mechanism..."** | \*\***"the definition as “certainty or utility” makes it hard to buy it..."** | NA | "Practical application of CTM model by CTM-AI..." | \*\***"Another concern is the relationship between the proposed framework and the CTM..."** | `Reviewer i9GU rebuttal`: We explain practical design choices regarding the up-tree and learning process.`Reviewer Wsj6 rebuttal`: We provide details about the motivation of chunk weights and implementation details of chunk generation. `Reviewer NwUJ rebuttal`: We clarify CTM-AI is a prototype inspired by the consciousness theory (CTM), **not** a model with consciousness. |

In addition, we expanded several minor discussions—such as the conceptual relationship between CTM-AI and multi-agent systems, Branish, post-training, and consciousness, as well as details on modality weighting and failure modes. We want to emphasize that these clarifications **do not affect the core contribution of the paper** and **are fully addressed in the rebuttal**.

---

> ### Author Response · Authors · 2025-12-03
>
> We emphasize two key points from the reviews and our rebuttal stage:
>
> **(1) Reviewers broadly agree on the strengths of the paper.**
>
> Across all reviews, there is a consistent consensus that the core idea is novel, the presentation is clear and easy to follow, and the empirical evaluation is solid and comprehensive. Such key strengths verify the core ideas of our CTM-AI paper.
>
> **(2) The remaining concerns are peripheral—not about the core contribution—and we have fully addressed them.**
>
> Most reviewer concerns focus on secondary aspects: (i) limited discussion of time/efficiency, (ii) absence of several multi-agent variant baselines, and (iii) insufficient detail on related concepts. These issues do not challenge the central contribution of our work: providing a blueprint for a general AI system, grounded in Conscious Turing Machine theory, and demonstrating a working system prototype. We have directly addressed all concerns during our rebuttal stage by adding:
>
> * Full results for multiple multi-agent baselines (adding baselines including multi-agent debate, query augmentation, multi-agent orchestration, ensemble voting)
> * Detailed efficiency and cost analysis (token usage, API-call cost, latency)
> * An additional **11 pages of technical details** in the Appendix for our revision (including prompt details, module-level implementation descriptions, algorithmic pseudocode, and extended case studies).
> * Additional discussions on the relationship between CTM-AI and CTM theory, multi-agent systems, Branish, post-training, and consciousness.

---

### Meta-Review · Area_Chair_HF2q · 2025-12-22

**Summary:**

The paper received five reviews, with initial scores of 2, 4, 6, 4, and 6.

Reviewer NwUJ gave the rating of 2. This reviewer raised many concerns: lack of technical details, weak connection between CTM-AI and CTM, unclear writing, and overclaim issue. The AC shares the same concern with this reviewer that the paper lacks many detaisl that make it difficult to properly evaluate the proposed components and reproduce the results. The rebuttal and revision do not fully address this concern. The AC predicts that this reviewer would remain negative about this work.

Reviewer JtxT gave the rating of 4. The main concerns include lack of evaluation on some parameters (i.e., lifetime T and height of binary tree h) and missing baselines. The rebuttal provided additional results which should have addressed most concerns except the evaluation on lifetime T. The new results show that the performance varies significantly with different T's and therefore the AC believes it is necessary to conduct experiments with more datasets (currently only MUsTARD) to evaluate this parameter. This would give practitioners a clearer guide on how to apply the proposed framework in practice. The AC predicts that this reviewer is more likely to keep the score of 4 unchanged after rebuttal.

Reviewer dnVs gave the rating of 6. This reviewer mentioned unfair evaluation because this work uses Gemini-2.0-flash-lite (2024) while the baselines are based on older models. The AC agrees that this is a valid concern. The rebuttal did not fully address this concern. The reviewer also requested inclusion of efficiency metrics, which were provided by in rebuttal. The AC predicts that this reviewer would maintain the score of 6.

Reviewer Wsj6 gave the rating of 4. This reviewer shared the same concern with NwUJ that the paper lacks many important details. The reviewer also requested efficiency metrics and comparison with post-training. Given that the revision still lacks important details for reproducing the work, the AC predicts that the reviewer would maintain the score of 4.

Reviewer i9GU gave the rating of 6. The main concerns seem to be caused by the lack of details and unclear writing as the reviewer questioned the distinction with multi-agent systems, discrepancy between theory and implementation, and ambiguous definition of the learning mechanism. The rebuttal did not fully address the writing problem. The AC predicts that the reviewer is likely to remain as 6.

Overall, the reviews are more negative and the weaknesses are salient. The AC recommends rejection.

**Reviewer Concerns:**

The AC believes the reviewers' concerns are half addressed only. The rebuttal has clarified some definition issues and provided additional results related to efficiency metrics and comparisons with more baselines. However, several key concerns remain unaddressed: 1) *Lack of technical details*. The revision provided more details in Appendix C & D but those details are insufficient to fully understand how the framework was implemented. This severely hinders the reproduction of the results in this work. It also makes it difficult to comprehensively evaluate the contributions. 2) *Evaluation on lifetime T*. The rebuttal provided evaluation of this parameter on MUsTARD. But the results raise more questions. Specifically, the performance varies significantly across different iterations, e.g., 68.99% with 1 iteration but 82.5% with 3 iterations. The paper needs to conduct further investigation into this parameter and discuss its impact on the whole system. 3) *Unfair evaluation*. The rebuttal explained that Gemini-2.0-flash-lite was also used as the base model via prompting but this is not convincing enough. The paper needs to compare with more advanced agentic workflows (and other foundation models).

**Reviewer Scores:**

The AC predicts that the final scores would be
- Reviewer NwUJ: 2 or 4
- Reviewer JtxT: 4
- Reviewer dnVs: 6
- Reviewer Wsj6: 4
- Reviewer i9GU: 6

Overall the reviews are more negative and there are many lingering concerns. The AC recommends rejection.

---

### Decision · Program_Chairs · 2026-01-26

Reject